# Designed optogenetic tool for bridging single-neuronal multimodal information in intact animals

Rong-Kun Tao [1,2,6] ✉, Le Sun[1,3,6], Yu Qian[1,3], Yi-Ming Huang[4], Yu-Han Chen [1,4], Chun-Yu Guan[5], Ming-Cang Wang[5], Yi-Di Sun [1] & Jiu-Lin Du [1,3,4] ✉

Integrating morphological, functional and molecular information of individual neurons is critical for classifying neuronal cell types and probing circuit mechanisms of brain functions. Despite the emergence of extensive single-neuronal morphology datasets largely via random sparse labeling, it remains challenging to map arbitrarily selected neuron's morphology in vivo, especially in conjunction with its functional and molecular characteristics. Here, we report a genetically encoded Photo-inducible single-cell labeling system (Pisces) that enables simple, rapid and long-term in vivo labeling of the entire morphology of arbitrary neurons, as exemplified in intact larval zebrafish. Pisces allows sequential tracing of multiple neurons within individual animals, facilitating brain-wide projectome mapping. Importantly, combined with in vivo calcium imaging, and fluorescence in situ hybridization or single-cell RNA sequencing, Pisces allows linking individual neurons' morphology characterization with their functional and/or gene expression investigation, respectively. This strategy promises to advance the construction of single-neuronal multimodal atlases and expedite the elucidation of neural circuitries underlying brain functions.

Brain functions are executed by highly specialized and intricately interconnected neural circuitries composed of diverse neuron types, each with distinct morphological, physiological, and molecular properties[1,2]. A comprehensive classification of neuron types and dissection of circuit mechanisms necessitate multimodal data integration at the single-cell level. In the past decade, the progress in genetically encoded calcium sensors and sequencing technologies has enabled the recording of neural activity and transcriptome profiling in population neurons in intact animals, respectively[2–4]. However, tracing the complete morphology of arbitrary individual neurons in intact animals, especially in a manner compatible with functional and molecular studies, remains a significant challenge[1,5].

Stochastic labeling methods, including sparse labeling and genetic mosaics, have enabled the visualization of individual neurons across brain regions for mesoscale projectome mapping and functional analysis of neural circuits[6,7]. However, these methods often require multiple injections and extensive screening to achieve specific labeling, and the stochastic nature limits the suitability for targeted neurons with no specific gene marker[7]. To enable precise neuron targeting, several optogenetic tools have been developed; however, the use in vivo was constrained by leakage and efficiency issues[6,8]. Conventional alternatives, such as microinjected dyes and photo-activatable or photoconvertible fluorescent proteins (PAFP or PCFP), allow targeted labeling of neurons in vivo[6,9,10], but the low diffusion

[1]Institute of Neuroscience, State Key Laboratory of Brain Cognition and Brain-Inspired Intelligence Technology, Center for Excellence in Brain Science and Intelligence Technology, Chinese Academy of Sciences, Shanghai, China. [2]Clinical Research Institute, The First Affiliated Hospital of Xiamen University, School of Medicine, Xiamen University, Fujian, China. [3]University of Chinese Academy of Sciences, Beijing, China. [4]School of Life Science and Technology, ShanghaiTech University, Shanghai, China. [5]Department of Anesthesiology, Taizhou Hospital of Zhejiang Province Affiliated to Wenzhou Medical University, Zhejiang, China. [6]These authors contributed equally: Rong-Kun Tao, Le Sun. ✉e-mail: taorongkun@xmu.edu.cn; forestdu@ion.ac.cn

speed from the soma (activated region) to long-projecting neurites impedes the complete visualization of the full neuronal morphology[11].

For obtaining multimodal information from single neurons, recent innovations in genetically encoded tools have integrated neuronal activity imaging with local neurite labeling (e.g., FuGIMA) or transcriptomic profiling (e.g., PhOTseq and CAMPARI)[12–15]. These methods are still insufficient for comprehensive, high-resolution tracing of full neuronal morphologies[12–15]. Thus, there is a critical need for a non-invasive single-neuron morphological tracing tool that is capable of labeling long-range axonal projections and compatible with single-neuron functional and molecular profiling in vivo[16–18].

In this study, we design a Photo-inducible single-cell labeling system (Pisces) by utilizing a nuclear chimeric protein composed of a photo-cleavable protein (PhoCl), a photoconvertible fluorescent protein (mMaple), and a balanced combination of nuclear localization signal (NLS) and nuclear export signal (NES). Upon PhoCl activation, the cleaved, photoconverted mMaple is actively transported by NES, enabling rapid tracing of the entire complex morphology of any single neuron in intact larval zebrafish. More importantly, Pisces is compatible with in vivo calcium imaging, single-cell RNA sequencing (scRNA-seq), and fluorescence in situ hybridization (FISH), and enables

the acquisition of multimodal information on individual neurons. This tool represents an advance in the study of mapping single-neuron axon projectomes, facilitating the construction of single-neuron multimodal atlases for the exploration of brain function mechanisms.

## Results

### Nuclear PhoCl allows single-cell activation in larval zebrafish

We proposed that precise single-cell manipulation in vivo can be achieved by nucleus-localized photo-cleavable protein PhoCl, which reduces the interference from nearby neuronal processes. Upon activation by violet light, PhoCl transitions from green to red fluorescence and ultimately becomes colorless through irreversible cleavage into two fragments[19] (Fig. 1a). This property enabled optogenetic applications with minimal background interference in cell cultures[19].

We expressed PhoCl and the photoconvertible protein Kaede[20] in the nucleus of larval zebrafish neurons using the pan-neuronal promoter *elavl3*, and evaluated their sensitivity to ambient light environments (ALE) at 6 days-post-fertilization (dpf) (Fig. 1a). PhoCl retained its green fluorescence under both dark and ALE in vivo, while Kaede was partially activated upon ALE. Both proteins could be activated under 405-nm LED illumination (Fig. 1b, c). Furthermore, nuclear PhoCl

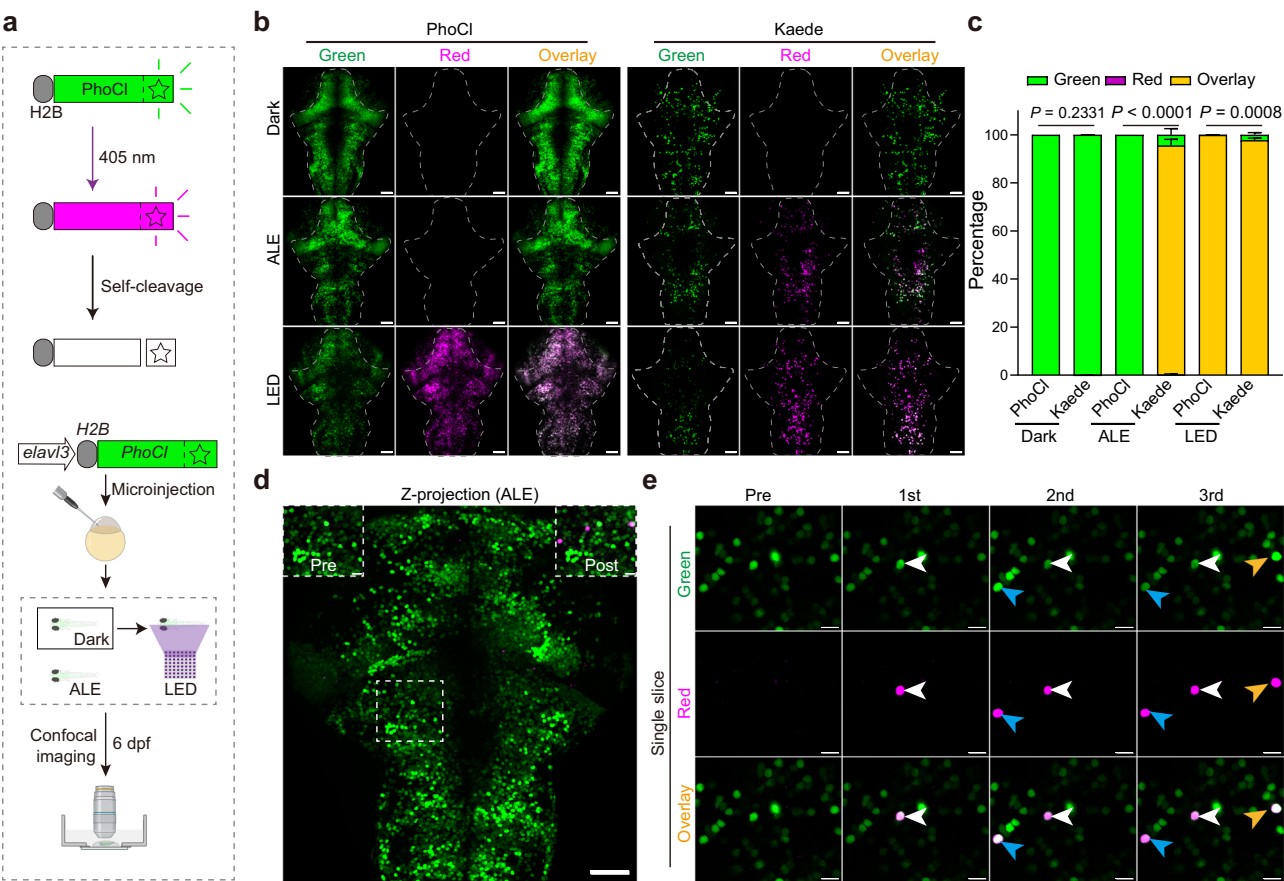

**Fig. 1 | Nuclear PhoCl allows single-cell manipulation in vivo. a** Schematic illustration showing the fluorescence detection of neuronal nuclear PhoCl in zebrafish at 6 days-post-fermentation (dpf). Two groups of larvae were raised under different conditions: one in constant darkness (Dark) and the other in an ambient light environment (ALE, ~250 lx). The fish raised in the dark were treated with 2 min 405-nm LED stimulation (0.2 mW/mm²). Activation of nuclear PhoCl converts its green fluorescence to red, which then becomes colorless following self-cleavage. Representative maximum projection images along the z-axis (**b**) and grouped analysis (**c**) of nuclear PhoCl and Kaede fluorescence signals in the brain (*n* = 7, 7, 10, 7, 12, 7 fish for each group accordingly). Scale bars: 50 μm. Some error bars in the grouped analysis are too small to be visible. The outline of the fish brain without

eyes is highlighted with a dashed line. Data are shown as mean ± s.e.m. Statistical significance was determined using a two-tailed Mann–Whitney U test (Dark and LED) or a Chi-squared test (ALE). **d** Representative z-axis maximum projection images showing H2B-PhoCl signals in the brain (*n* = 7 fish). The dashed square indicates the region for activation, with pre-activation (Pre) and post-activation (Post) images shown in the upper left and upper right corners, respectively. Scale bars: 50 μm in whole images, 10 μm in zoomed views. **e** Single-slice fluorescence image demonstrating the sequential activation of three neurons expressing nuclear PhoCl under 405-nm laser (0.5 μW) for 10 s. Neuron 1 is marked with a white arrowhead, neuron 2 with a blue arrowhead, and neuron 3 with an orange arrowhead. Scale bars: 10 μm (*n* = 7 fish). See also Supplementary Fig. 1.

could be selectively activated in a specific field of view or sequentially activated in arbitrary single neurons with a 405-nm laser, while remaining resistant to wavelengths ranging from 445-nm to 640-nm (Fig. 1d, e and Supplementary Fig. 1a, b). Besides, nuclear PhoCl did not obviously interfere with calcium activities in either cultured HEK293 cells or neurons in larval zebrafish (Supplementary Fig. 1c–j). We did not test the properties of nuclear Kaede in these experiments. These results demonstrate that nuclear PhoCl enables single-cell manipulation in vivo under dark and even ambient light conditions.

### Design a nuclear PhoCl-based, light-controllable protein translocation tool for labeling individual neurons in vivo

PhoCl has been previously reported to manipulate protein translocation between the cytosol and nucleus through free diffusion in cultured cells[19,21]. We then developed a light-controllable nucleus-to-cytosol protein translocation system with a rapid protein trafficking rate by fusing PhoCl with the red fluorescence protein mCherry, and different versions of NLS and NES (Supplementary Fig. 2a). We examined the translocation efficiency of the 35 combinatory constructs in H1299 cell cultures using a custom-made LED setup (Supplementary Fig. 2b, c). The construct, termed Pisces0.1, containing duplex NES and multiple NLS flanking the chimeric mCherry and PhoCl2c, displayed strong nuclear localization under control conditions and showed the

highest cytosolic translocation efficiency upon bulk activation (Supplementary Fig. 2c and Supplementary Note 1). Under 2-min 405-nm laser stimulation, Pisces0.1 was activated, causing a rapid decrease in nuclear green fluorescence and a significant increase in the cytosolic mCherry signal (Fig. 2a, b and Supplementary Fig. 2d–h). This resulted in approximately a 4-fold increase in the cytosolic localization ratio within 30 min (Supplementary Fig. 2i).

In larval zebrafish expressing Pisces0.1, the majority of neurons showed accurate nuclear localization, with minor cytosolic leakage observed in a small subset of cells (Fig. 2c, d, blue arrowheads in zoomed-in view, and Supplementary Movie 1). After LED illumination, neuronal processes became visible in most neurons; however, some neurons exhibited ineffective morphological labeling (Fig. 2d, white arrowheads in zoomed-in view, and Supplementary Movie 1). The minor cytosolic leakage and ineffective labeling are possibly due to the aggregation tendency of mCherry[22,23]. In addition, the high baseline mCherry signals in non-activated neurons could reduce contrast and clarity in single-neuron morphology labeling experiments.

To optimize the system, we created an improved version called Pisces by replacing mCherry with the PCFP mMaple (Fig. 2c and Supplementary Note 1), which is a high-performance, monomeric, green-to-red photoconvertible fluorescent protein optimized for multimodal imaging applications and demonstrated superior photostability,

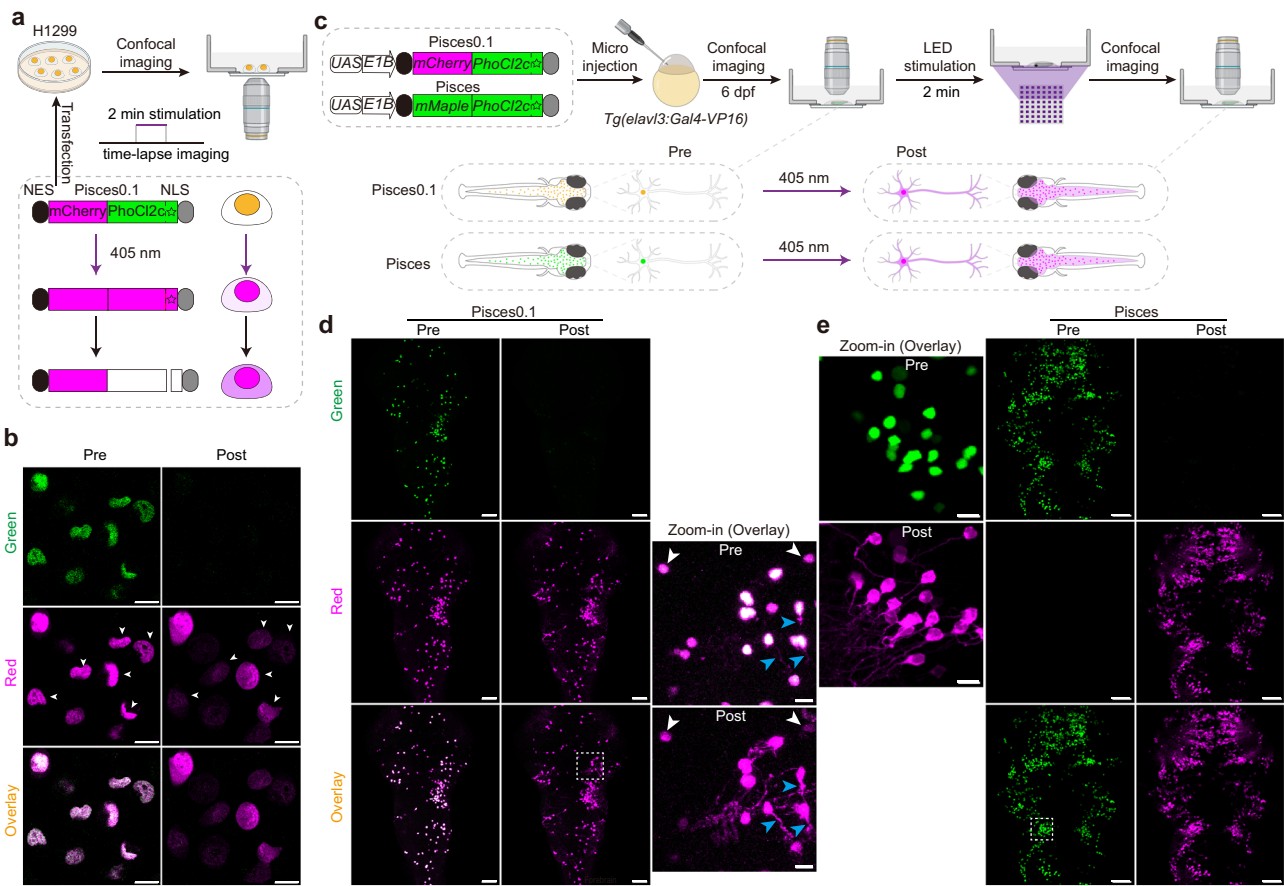

**Fig. 2 | In vitro and in vivo characterization of a PhoCl-based single-cell labeling system. a** Schematic depicting Pisces0.1 characterization in H1299 cell cultures. Pisces0.1 shows both green and red fluorescence in the nucleus. The green fluorescence vanishes after 2 min of continuous 405-nm laser activation (7.5 μW), followed by PhoCl translocation to the cytosol after self-cleavage. **b** Fluorescence images of H1299 cells expressing Pisces0.1 before and after 405-nm laser illumination. Scale bars: 10 μm ($n = 4$ imaging experiments). Fluorescence traces of six cells (indicated by white arrowheads) can be found in Supplementary Fig. 2d–f. **c** Schematic of Pisces0.1 and Pisces characterization in 6-dpf larval zebrafish using

bulk activation with a custom LED setup (1.4 mW/mm², 2 min). Representative images of zebrafish larvae expressing neuronal Pisces0.1 (**d**) and Pisces (**e**). Zoomed-in regions (dashed rectangle) of individual neurons are highlighted. Neuronal Pisces0.1 shows cytosolic leakage (blue arrowheads) before activation and insufficient translocation after activation (white arrowheads). In contrast, all Pisces-positive neurons display proper nuclear localization and complete translocation after activation. Larvae were raised in constant darkness. Scale bars: 50 μm in whole images, 10 μm in zoomed views ($n = 6$ fish). See also Supplementary Fig. 2, Supplementary Note 1 and Supplementary Movie 1.

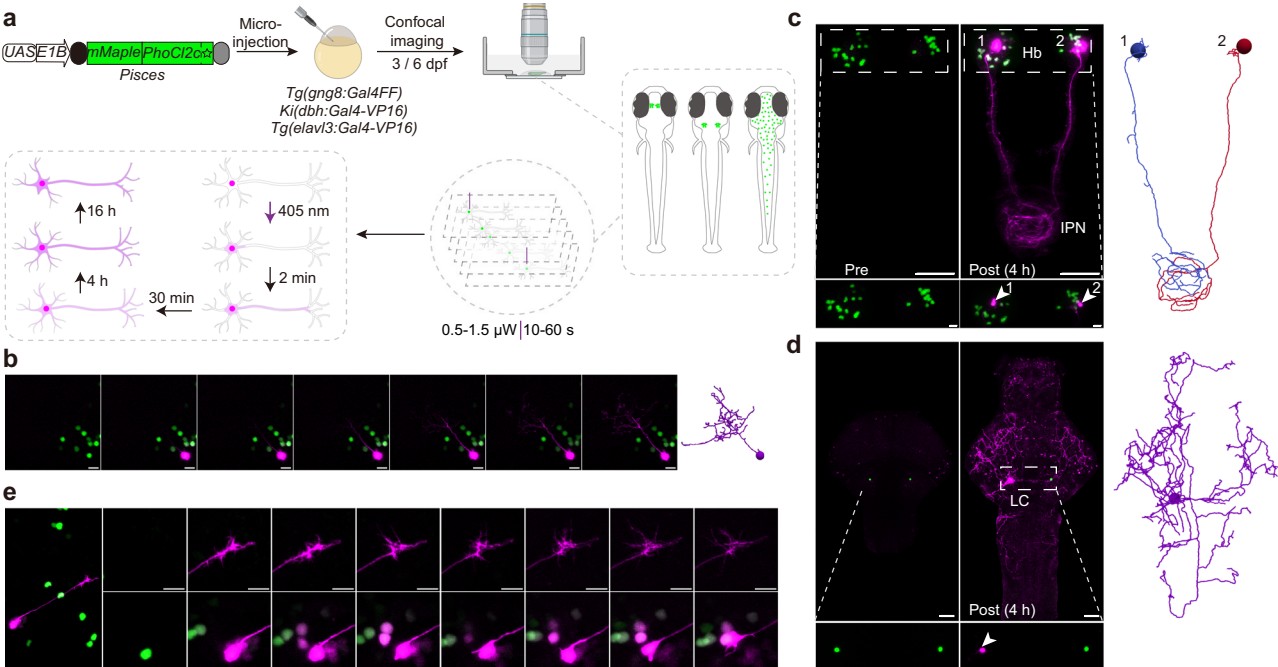

**Fig. 3 | In vivo characterization of Pisces for tracing single-neuron morphology. a** Schematic representation of Pisces expression in the habenula, locus coeruleus (LC), and the whole brain of 3 or 6-dpf larval zebrafish. Single pulses of a 405-nm laser (0.5 µW for 60 s or 1.5 µW for 10 s) were used for activation. **b** Time-lapse imaging demonstrates the rapid axonal trafficking of Pisces in tectal neurons following 405-nm laser activation (1.5 µW for 10 s) on 6-dpf larval zebrafish. White arrowheads show the axonal progression of Pisces labeling, while blue arrowheads mark the activated neuron nucleus. Pisces labels the entire axon within 2 min, and all processes are labeled within 30 min. The estimated axon trafficking rate is 1.02 ± 0.06 µm/s based on 14 neurons. The morphological trace of an activated neuron is shown on the right. Scale bars: 10 µm. Larvae were raised in constant darkness. Representative z-axis maximum projection images of the morphological projections of two activated habenula neurons on both sides (**c**) and one LC-NE

neuron (**d**) on 6-dpf larval zebrafish. The brightness of red fluorescence was adjusted to visualize the nuclei of activated neurons in the lower panels. Numbers and white arrowheads indicate the activated neurons. Morphological traces of each neuron are displayed on the right (*n* = 3 fish). Scale bars: 50 µm in whole images, 10 µm in zoomed views. Hb habenula, IPN interpeduncular nucleus. Larvae were raised in constant darkness. **e** Fluorescent images showing tectal neuron morphology (left, at 2 h post-activation), and neurites and filopodia (right) dynamics of a tectal neuron in a 3-dpf zebrafish over 14 h. White arrowheads indicate neurite changes. Laser activation was performed with 405-nm (0.5 µW for 60 s) (*n* = 6 fish). Scale bars: 10 µm. Raw images and traces can be found in Supplementary Fig. 4c. Larvae were raised in ALE. See also Supplementary Figs. 3–5 and Supplementary Movies 2–6.

enhanced folding efficiency, and excellent performance in super-resolution microscopy when compared with other commonly used PCFPs (such as mEos2 and mClavGR2)[24]. All neurons expressing Pisces showed strong green nuclear fluorescence without red background signal prior to activation (Fig. 2e). After LED illumination, the green fluorescence in the nucleus completely disappeared, and intense red fluorescence filled the cytosol in all activated neurons, allowing for the visualization of individual neuronal morphology (Fig. 2e and Supplementary Movie 1). To clarify the working principle, we generated a schematic figure (Supplementary Fig. 2j) that compares the expected localization and photoconversion outcomes for a PCFP with an NLS, NES, without any targeting sequence, Pisces0.1, and the final Pisces construct.

### Pisces enables fast, complete, and stable labeling of arbitrary neurons in larval zebrafish

To achieve high-resolution in vivo single-neuron morphology tracing, we characterized the properties of Pisces in zebrafish (Fig. 3a). We found that a single 10-s pulse of 405-nm laser activation successfully labeled the morphology of neurons in the optic tectum (OT) with neurite labeling progressing at a rate of 1.02 ± 0.06 µm/s in 6-dpf larval zebrafish (Fig. 3b), which aligns with the rate of fast components of axonal transport[11].

To further demonstrate the capability of Pisces in outlining the complete morphology of targeted neurons, we expressed it in habenular (Hb) neurons in the *Tg(gng8:Gal4FF)* fish line and these neurons send long-range axon projections to the interpeduncular nucleus (IPN)

located in the ventral midbrain[25,26] (Fig. 3a and Supplementary Fig. 3a). Following precise activation of single neurons on one or both sides of the habenula, the axon terminals in the IPN became visible within 4 h, under both dark and ambient light conditions (Fig. 3c, Supplementary Fig. 3b, c and Supplementary Movie 2). We further validated the morphological completeness of Pisces signals by performing single-cell electroporation of Alexa Fluor 488 dye into Pisces-activated neurons in *Tg(elavl3:Gal4-VP16);Tg(UAS-E1B:Pisces)* zebrafish larvae, which exhibited sparse Pisces expression (Supplementary Fig. 4a, b and Supplementary Movie 3). This experiment revealed colocalization of Pisces (red) and Alexa Fluor 488 (green) signals along axonal paths and terminals, demonstrating Pisces' reliability in tracing the complete morphology of individual neurons.

We also examined the performance of Pisces in locus coeruleus (LC) norepinephrine (NE) neurons (Fig. 3a), known for their complex brain-wide axon projections[27]. After activating the LC-NE neuron in the brainstem of *Ki(dbh:Gal4-VP16)* fishline[28], Pisces labeled the extensive axon projections throughout the brain within 4 h (Fig. 3d and Supplementary Movie 4). This result indicates that Pisces can rapidly label complete and complex neuronal morphologies in vivo. The rapid diffusion of the fluorescent signal of Pisces throughout the neuron is facilitated by the active transport of NES, ensuring comprehensive cytosolic labeling.

To assess the ability of Pisces to label the morphology of neurons in ventral brain regions, we targeted neurons with somata located in deeper regions (-80–220 µm below the dorsal surface). Pisces is capable of labeling and reconstructing individual neurons in these ventral

areas (Supplementary Fig. 4c and Supplementary Movie 5). However, the fluorescence intensity and signal-to-noise ratio (SNR) of ventral neurons were lower than those of dorsal neurons, primarily due to optical scattering and light attenuation at greater imaging depths, a known limitation of confocal microscopy in thick tissues.

Tracing the morphology of individual neurons in zebrafish with dense Pisces expression presents a significant challenge. To evaluate Pisces' ability to label neuronal morphology in larvae with varying densities of Pisces-positive neurons, we conducted neuronal labeling in the *Tg(elavl3:Gal4-VP16);Tg(UAS-E1B:Pisces)* larvae, which exhibited different labeling densities in the larval zebrafish brain. With careful experimental handling (see "Methods" section), Pisces effectively labeled and reconstructed individual neuronal morphologies, even in brains with high density of Pisces expression, reaching up to 40% (Supplementary Fig. 4d, e and Supplementary Movie 6).

Pisces also exhibited stability in long-term neuronal labeling. Fluorescence signals in labeled projections remained stable for over 16 h (Supplementary Fig. 5a, b), making it suitable for tracking dynamic changes in neuronal processes over extended periods. As shown in 3-dpf zebrafish, Pisces enabled observation of dynamic extensions and retractions of neurites and filopodia over 14 h (Fig. 3e, Supplementary Fig. 5c and Supplementary Movie 7), suggesting that Pisces can be used to examine the development of neuronal processes[29].

Taken together, these results highlight Pisces as a powerful and user-friendly tool for in vivo single-neuron morphology labeling, with broad utility for studies of neural development.

## Pisces enables sequential labeling of adjacent and separated neurons in the nervous system

Mapping the whole-brain projectome requires resolving the complex organization of densely packed neurons with diverse and long-range projections, which are critical for understanding the intricate network of neuronal connections underlying brain function[30]. A key challenge is how to distinguish adjacent neurons with high specificity, as their morphology may intermingle[31]. To address this issue, we conducted sequential activation experiments using Pisces (Fig. 4a). We showed that the targeted neurons were sequentially labeled with high-resolution (Fig. 4a–c and Supplementary Movie 8), while the limited photoconversion of mMaple in the neighboring neuron without PhoCl activation ensured labeling specificity (Fig. 4b, zoomed-in view). This result demonstrates its capacity for snapshotting adjacent neurons with exceptional specificity.

Visualizing multiple neurons' morphology in a single animal can speed up projectome mapping. By activating multiple neurons across the brain and spinal cord, Pisces achieved clear labeling of nine individual neurons with extensive axonal projections, one of which spans approximately 1 mm across the spinal cord (Fig. 4d, e and Supplementary Movie 9). These findings indicate that Pisces acts as a robust and versatile tool for high-resolution, efficient, and specific mapping of the brain-wide axonal projectome.

## Pisces permits mapping the morphology of arbitrary neurons after functional characterization in combination with in vivo calcium imaging

A currently common strategy for studying neural mechanisms underlying brain functions is to perform large-scale in vivo optical calcium imaging to monitor the activities of population neurons. Following functional characterization, it is essential to further examine the morphology of these functionally identified neurons within intact animals[12,13]. To evaluate the suitability of Pisces for this function - morphology integration analysis, we focused on the optic tectum (OT), a brain region composed of periventricular neurons (PVNs) with diverse functional properties. The heterogeneity of PVNs presents challenges in correlating their functional profiles with anatomical substrates[32,33]. While recently developed FuGIMA has well linked the

visual response properties of PVNs to their dendritic arborization patterns[12,13], obtaining a comprehensive view of the entire morphological architecture of individual PVNs remains an unmet need. To address this, we expressed nuclear Pisces in the *Tg(elavl3:GCaMP6s)* fish line, which allows us to monitor neuronal activity in response to visual stimuli using cytosolic GCaMP6s (Fig. 5a). Following the functional assay, we activated nuclear Pisces within the characterized neurons to trace their complete morphology (Fig. 5a).

Using this strategy, we successfully traced, reconstructed, and registered the morphology of 131 functionally characterized OT neurons, including 114 PVNs and 17 non-PVNs, to a standardized brain coordinate template (Supplementary Fig. 6a). Functionally, these neurons were classified into five functional categories based on their light response profiles: transient ON (tON), sustained ON (sON), transient OFF (tOFF), sustained OFF (sOFF), and ON-OFF (Fig. 5b and Supplementary Movies 10, 11). Morphologically, these PVNs were further subdivided into periventricular interneurons (PVINs) and periventricular projection neurons (PVPNs) based on whether their axons extended beyond the OT neuropil (Fig. 5c). Interestingly, PVINs and PVPNs displayed significant differences in their constitution across light response types, especially the sON and sOFF categories (Fig. 5d). Process distribution analysis within specific OT neuropil layers revealed differences in process abundance between PVINs and PVPNs (Supplementary Fig. 6b). PVPNs exhibited a higher process density within the stratum album centrale (SAC)/stratum periventriculare (SPV) layer (Fig. 5e), consistent with the SAC/SPV layer's role as the primary output channel of the OT[32].

Functional analysis revealed distinct light response patterns in PVNs that exhibited relative layer specificity across the OT neuropil (Supplementary Fig. 6c). ON-responsive PVNs predominantly innervated the superficial stratum fibrosum et griseum superficiale 1-2 (SFGS1-2) layers, while PVNs with OFF responses were more frequently localized to the deeper stratum griseum centrale (SGC) layers (Supplementary Fig. 6d, e). This property likely arises from direct retinotectal inputs, as OFF-responsive axons from retinal ganglion cells preferentially project to deeper layers, subserving rapid behavioral responses to environmental changes[34].

Furthermore, we observed notable differences in light responses between PVPNs with contralateral and ipsilateral descending tectofugal projections. PVPNs with contralateral projections exhibited a higher prevalence of ON-OFF responses, whereas those with ipsilateral projections were characterized by sON and tOFF responses (Fig. 5f, g). Contralateral descending PVPNs also displayed a wider range of light response types, more extensive dendritic arborization, and broader tectofugal projection patterns (Fig. 5g, h). Previous reports show that ipsilateral descending projections are tied to threat avoidance, while contralateral projections are associated with prey approach in vertebrates[35,36]. Our findings imply that predation behaviors may involve a more intricate visuomotor transformation process, characterized by a greater diversity of light response types, expanded dendritic structures, and more extensive tectorecipient regions[37].

Overall, these results demonstrate that Pisces enables effective integration of single-neuron morphological tracing with functional characterization, offering a comprehensive framework to elucidate the anatomical underpinnings of functional wiring diagrams.

## Pisces allows reliable isolation of photoconverted neurons for subsequent transcriptional profiling in combination with scRNA-seq

Molecular markers serve as invaluable tools for labeling and analyzing neuronal cell types[1], but methods to correlate molecular markers with neuronal function and morphology remain underdeveloped. To explore the feasibility of isolating Pisces-activated GCaMP6s-positive neurons, we selectively activated individual neurons in 6-dpf larval

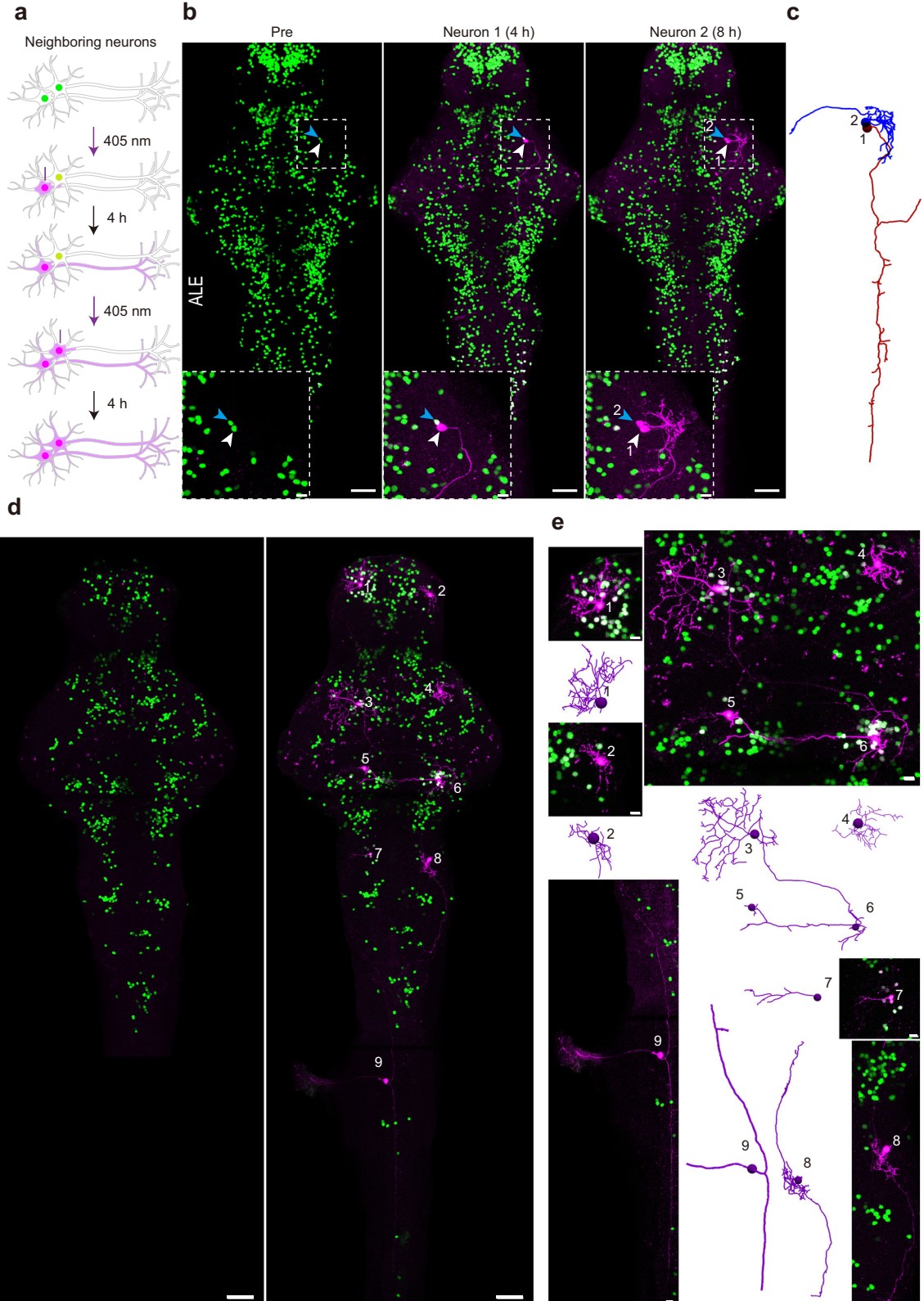

**Fig. 4 | Pisces enables tracing the morphology of adjacent neurons and separated neurons across the brain. a** Schematic illustrating the sequential activation of two neighboring neurons in the midbrain of 6-dpf larval zebrafish.
**b**, **c** Fluorescent images showing the morphology of two tectal neurons activated sequentially. The first neuron ("1") was designated by a white arrowhead, and the second ("2") by a blue arrowhead (**b**). The dashed box highlights two sequentially activated neurons, shown in detail in the zoomed-in view below. Morphological traces of these two adjacent neurons are displayed in (**c**). Larvae were raised under

ALE conditions. Scale bars: 50 μm in whole images, 10 μm in zoomed views. *n* = 6 fish. Representative z-axis maximum projection images (**d**), corresponding zoomed-in views, and morphological tracing (**e**) of nine neurons activated across the entire brain on 6-dpf larval zebrafish. Numbers indicate the activated neurons, which are shown in higher magnification in the zoomed-in panels on the right (*n* = 3 fish). Note that neuron 9 is not included in the pre-activation image. Larvae were raised in constant darkness. Scale bars: 50 μm in whole images, 10 μm in zoomed views. See also Supplementary Fig. 5, Supplementary Movies 7–9.

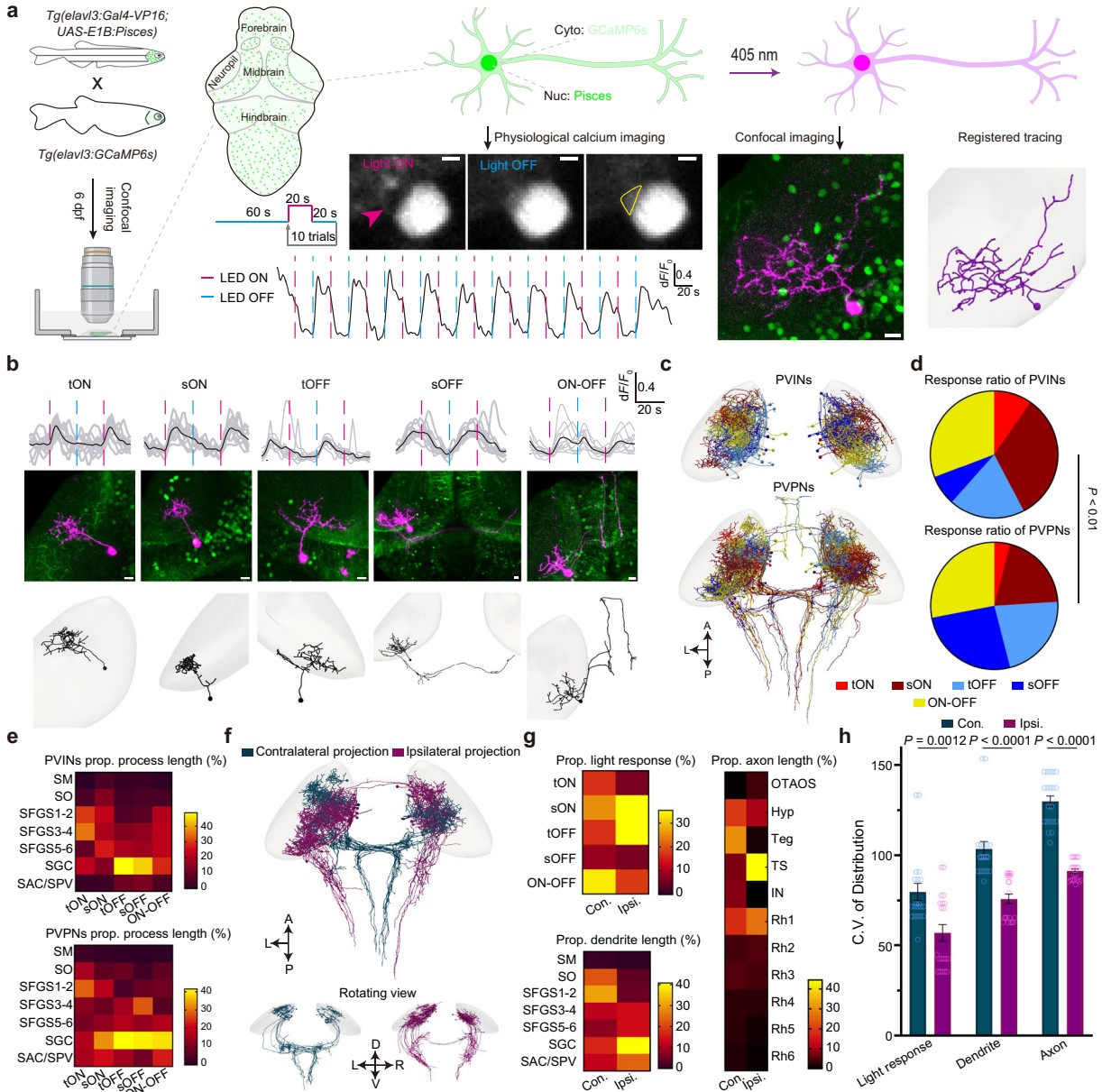

**Fig. 5 | Integrative characterization of morphology and function of individual tectal neurons in vivo. a** Schematic of the experimental design combining light-induced calcium imaging and morphological reconstruction of tectal neurons. Signals in the peri-nuclear soma region were used for functional measurements (scale bars: 2 μm). Morphological tracing via nuclear Pisces activation (scale bars: 10 μm). Reconstructions registered to the brain template. Neuropil is shaded in gray. **b** Five distinct neuron types were categorized by aligning average calcium response traces with light stimulation patterns. Representative traces (top), neuronal morphologies (middle), and registered reconstructions (bottom) are shown (scale bar: 10 μm). Types include tON ($n = 7$), sON ($n = 27$), tOFF ($n = 21$), sOFF ($n = 17$), and ON-OFF ($n = 29$). **c** Collection of all periventricular interneurons (PVIN, $n = 56$) and periventricular projection neurons (PVPN, $n = 58$), registered to the brain template. **d** Pie charts showing neuron-type proportions within the PVIN (top) and PVPN (bottom) populations. Statistical significance was determined using a Chi-squared test. Detailed $n$ numbers are in Source Data. **e** Proportions (prop.) of

process length for PVIN (top) and PVPN (bottom) neurons across different neuropil layers. SM stratum marginale, SO stratum opticum, SFGS stratum fibrosum et griseum superficiale, SGC stratum griseum centrale, SAC stratum album centrale, SPV stratum periventriculare. **f** Collection of all contralateral ($n = 13$) and ipsilateral ($n = 21$) descending projecting PVPN neurons, registered to the brain template. Note that a minority of ipsilateral neurons ($n = 2$) project across the midline. Proportional statistics (**g**) and coefficient of variation analysis (**h**) comparing luminance response, dendrite length, and axon length distribution between contralateral (con.) and ipsilateral (ips.) projecting PVPN neurons ($n = 20$). OTAOS optic tract and accessory optic system, Hyp hypothalamus, Teg tegmentum, TS torus semicircularis, IN interpeduncular nucleus, Rh rhombomere (numbers denote different segments). Error bars represent s.e.m. Statistical significance was determined using a Chi-squared test. See also Supplementary Fig. 6, Supplementary Movies 10 and 11.

zebrafish within distinct brain regions, including the left habenula (lHb), right habenula (rHb), and midbrain, followed by dissection and fluorescence-activated cell sorting (FACS) (Fig. 6a). The photo-activated neurons displayed strong red fluorescence, enabling precise identification and efficient sorting (Supplementary Fig. 7a, b).

To access data quality, we compared the transcriptome profiles from lHb and rHb neurons with a previously reported whole Hb dataset[38], confirming a high consistency between the two datasets (Supplementary Fig. 7c). Differential gene expression (DGE) analysis of habenular and midbrain neurons (total number: 202) identified a set of

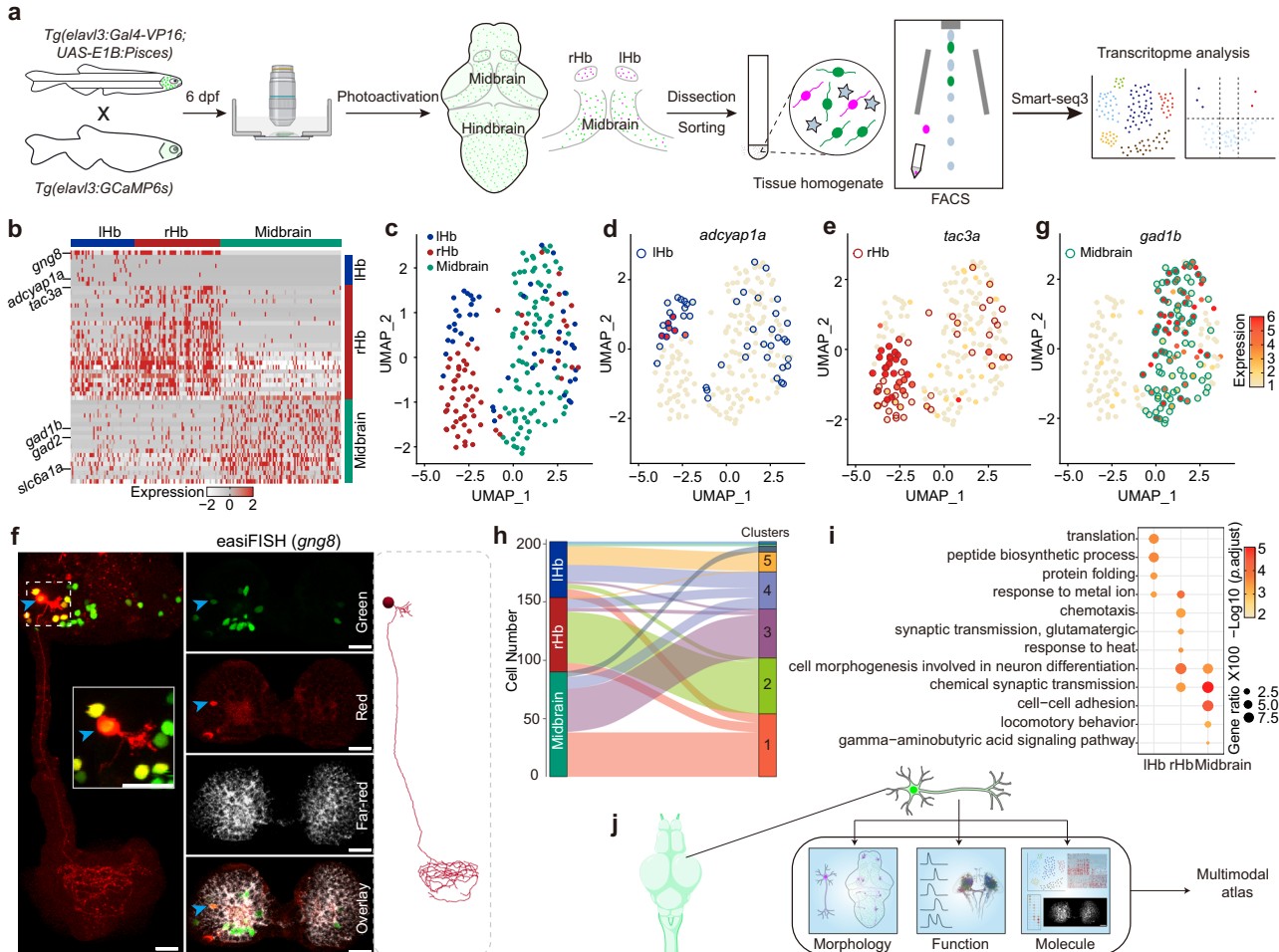

**Fig. 6 | Transcriptome analysis of Pisces-activated GCaMP6s-positive neurons. a** Overview of the workflow for isolating and analyzing single-cell transcriptomes from Pisces-activated, GCaMP6s-positive neurons in the left habenula (lHb), right habenula (rHb), and midbrain. **b** Heatmap showing differentially expressed genes across lHb, rHb, and midbrain neurons. Marker genes include *gng8* (habenula-specific), *adcyap1a* (lHb), *tac3a* (rHb), *gad1b, gad2, and slc6a1a* (midbrain). **c** Uniform manifold approximation and projection (UMAP) clustering of neurons from lHb ($n = 48$), rHb ($n = 64$), and midbrain ($n = 90$) identify three distinct populations. Distribution patterns of habenular neurons expressing enriched marker genes in the UMAP clusters, including *adcyap1a* in the lHb (**d**) and *tac3a* in the rHb (**e**). **f** Representative z-axis maximum projection images of all 211 slices illustrating the integration of *gng8* mRNA spatial localization and single habenular neuron morphology in zebrafish larvae expressing neuronal Pisces. Left: Morphology of individual habenular neurons, imaging before easiFISH with a zoomed-in view of the activated neuron (dashed square). Middle: Projection images of 10 slices showing gng8 mRNA expression detected by easiFISH. Nuclear-localized Pisces, activated neurons, and mRNA signals are shown in green, red, and far-red (white), respectively. After easiFISH, neurite signals diminish, leaving only nuclear fluorescence. Right: Morphological reconstructions of neurons before easiFISH. Activated neuron indicated by blue arrows. Similar results were observed from 3 larvae (ALE conditions). Scale bars: 20 μm. **g** Distribution patterns of midbrain neurons expressing enriched marker genes *gad1b* in the UMAP clusters. **h** Sankey plot illustrating the relationships between known lHb, rHb, and midbrain neurons and eight unsupervised cell clusters identified in the unbiased UMAP analysis. Clusters 1–5 are labeled; clusters 6–8 are included but unlabeled due to space limitations. Mutual information (MI = 0.50) indicates the distinct characteristics of each cell type. **i** Gene Ontology (GO) enrichment analysis for neurons from the three brain regions. Statistics were performed using a two-sided Hypergeometric test with Benjamini−Hochberg correction for multiple comparisons. Summary model integrating molecular, morphological, and functional data from single neurons across the whole brain in vivo using the Pisces system. See also Supplementary Fig. 7, Supplementary Movies 12 and 13.

enriched marker genes (Fig. 6b) and distinct neuronal subpopulations (Fig. 6c). Notably, the habenular marker gene *gng8*, along with established asymmetric markers *adcyap1a* for the lHb and *tac3a* for the rHb[38], were also observed within our dataset (Fig. 6d, e and Supplementary Fig. 7d). To address the gap toward morphology-transcription mapping, we performed a customized multiplexed fluorescence in situ hybridization (easiFISH) on Pisces-labeled individual habenular neurons in Tg(*elavl3:Gal4-VP16*);Tg(*UAS-E1B:Pisces*) larvae. Traced neurons from two groups of fish were probed for *gng8* and *tac3a* mRNA, respectively. Most habenular neurons expressed *gng8* mRNA, while *tac3a* mRNA was predominantly restricted to right-sided habenular neurons (Fig. 6f, Supplementary Fig. 7e and Supplementary Movies 12, 13), consistent with previous findings[38]. These results suggest a single-neuron level correlation between morphotype and molecular identity.

The transcriptomic analysis of midbrain neurons revealed a predominantly GABAergic profile, characterized by high expression of *gad1b, gad2* (Glutamate Decarboxylase), and *slc6a1a* (GABA Transporter 1) (Fig. 6g and Supplementary Fig. 7f, g), consistent with previous studies[37,39]. Moreover, unbiased UMAP analysis identified eight distinct subclusters among the neurons with known locations, suggesting higher cell heterogeneity within habenular neurons (especially for lHb) compared to midbrain neurons (Fig. 6h and Supplementary Fig. 7h).

Gene Ontology (GO) analysis further revealed unique biological pathways enriched in specific brain regions (Fig. 6i). The rHb neurons were enriched in pathways related to chemotaxis, heat response, and glutamatergic signaling, while lHb neurons showed pathways associated with peptide biosynthesis and protein folding (Fig. 6i),

supporting previously noted lateralization and developmental differences between the left and right habenula[40,41]. Midbrain neurons were enriched in pathways involving cell-cell adhesion, locomotory behavior, and GABA signaling (Fig. 6i), consistent with their known functions[42].

These results demonstrate that Pisces-activated GCaMP6s-positive neurons can be effectively sorted for transcriptome profiling, enabling a multimodal approach to integrate molecular identity with neuronal function and morphology (Fig. 6j).

## Discussion

Our study presents Pisces as a well-designed light-controllable protein translocation tool that enables efficient in vivo tracing of single-neuron morphology in the intact larval zebrafish, even under an ambient light environment, a common setting in neuroscience experiments. Unlike traditional genetically encoded methods (see Supplementary Fig. 2j), which exhibit limited effectiveness in ambient light conditions, require repeated activation in the soma region, and take days to weeks for complete labeling due to slow protein diffusion rate ($\sim$0.02–0.09 μm/s)[11], Pisces achieves precise manipulation by nuclear localization and rapid active, energy-dependent nuclear export by NES ($\sim$1 μm/s)[43] (Supplementary Table 1). This enables complete and high-resolution single-neuron morphological labeling within 4 h in larval zebrafish.

Whole-brain axonal projectome, a comprehensive map detailing the morphological trajectory of individual neurons across the brain, is typically generated by using time-intensive and technically demanding sparse labeling[9,30,44]. However, these conventional approaches often result in either over-labeling or under-labeling due to the random nature of the labeling[7]. In the current study, we demonstrate that Pisces enables high-resolution and clear tracing of the complete morphology of multiple neurons, including adjacent neurons and even the types with highly complex axonal arborization, such as LC-NE neurons. This ability would facilitate comprehensive whole-brain projectome mapping, aiding in neuronal type classification and functional prediction based on axonal projections. While Pisces is effective in larval zebrafish, it deserves to test its compatibility in other small animal models in the future. It is noteworthy, however, that Pisces is ineffective under two-photon activation, likely due to low activation efficiency, as has been reported with similar tools like PAmCherry and CAMPARI[14,15].

The labeling stability over long-term imaging sessions makes Pisces especially valuable for investigating neural development. In this study, we conducted a preliminary test on the dynamics of neurites and filopodia in tectal neurons at 3-dpf larvae. The high resolution and stability of Pisces would enable an in-depth study of early neuronal development processes, including proliferation, differentiation, migration, and axon guidance[45]. Given the conservation of neural developmental programs across vertebrates[46], Pisces holds significant potential for applications in developmental neuroscience, such as the construct of a comprehensive neuronal developmental landscape to link morphological dynamics with molecular information.

Large-scale in vivo optical calcium imaging is currently an important way to monitor the activities of population neurons in intact animals. One of the next bottlenecks is how to examine the morphology of functionally characterized neurons for probing neural circuitries involved. Pisces effectively addressed this longstanding critical limitation in the morphological tracing of arbitrary neurons, as demonstrated by mapping tectal neurons' complete morphology after calcium imaging. We revealed the distinct properties between PVINs and PVPNs based on integrative analysis of both the light response types and neuronal process distributions within specific neuropil layers, and these properties are correlated with the roles of these neurons in visuomotor functions. Moreover, we provided unprecedented insights into the functional neuroanatomy of contralateral and ipsilateral PVPNs, which are involved in two types of innate behaviors[35,36].

The combination of Pisces and in vivo calcium imaging would help to elucidate neural circuitries underlying behaviors.

Single-cell transcriptomics provides critical molecular information for cell type classification but lacks spatial context and may miss rare neuron types[47,48]. We demonstrated that Pisces-labeled neurons can be sorted out for transcriptomic analysis, even co-expression with calcium sensor, and revealed molecular heterogeneity among neuron types, suggesting the specific function of different brain regions, such as lateralization of the habenula[38]. Moreover, we also demonstrated that Pisces can be compatible with fluorescence in situ hybridization to integrate mRNA spatial pattern with morphology[4,49].

This study establishes Pisces as a powerful and compatible tool for integrating morphological, functional, and molecular data at the single-cell level within an intact organism. This system can be optimized by co-expression of Pisces with calcium sensors in a different spectral range, such as infrared[50,51], to facilitate in vivo calcium imaging and enhance contrast for cell sorting. We expect that integrating Pisces with other technologies could facilitate more comprehensive multimodal analyses within the same single neurons across the whole-brain scale.

## Methods

### Ethical Statement

Zebrafish were maintained following standard procedures. The experimental protocols involving zebrafish were approved by the Chinese Academy of Sciences (Approval no. NA-046–2019).

### Plasmid constructions

To compare the fluorescent properties of PhoCl and Kaede, we amplified the cDNAs of PhoCl from *pcNDA-NES-PhoCl-mCherry* (Addgene plasmid no. #87690) and Kaede from *pminiTol2-flk1-Kaede*. Both were subsequently subcloned into the pcDNA3.1-H2B vector for nuclear localization. For expression in zebrafish neurons, H2B-PhoCl and H2B-Kaede genes were amplified and ligated into the pminiTol2 vector harboring the *elavl3* promoter. The jRGECO1a gene was cloned from *pGP-CMV-NES-jRGECO1a* (Addgene plasmid no. #61563) and inserted into the pcDNA3 vector containing a duplicated nucleus-excluded signal peptide (NES) derived from MAPKK[52].

To develop a PhoCl-based light-inducible protein translocation system, we assembled various NES, nuclear localization signal peptides (NLS), PhoCl, and mCherry into the pcDNA3.1 vector. The amino acids of different NES and NLS are as follows. NES1: LQNELALKLAGLDINKTGGGS; NES2: LPPLERLTL; NES3: SELQNKLEELDLDSYK; NES4: ALQKKLEELELDE; NLS (c-myc): PAAKRVKLD. Multiple NES or NLS sequences are assembled consecutively without linkers.

We optimized translocation efficiency by substituting PhoCl with PhoCl2c and adjusting the linker lengths between PhoCl2c and the NES or NLS. These optimizations resulted in the construct dNES4-mCherry-PhoCl2c-6myc-H2B, termed as the Photo-inducible single-cell labeling system (Pisces0.1). The cDNA of Pisces0.1 was cloned into the *pminiTol2-10UAS-E1B* vector for expression in zebrafish. The following oligonucleotides were synthesized by TsingkeBio and used for the construction of Pisces0.1 and Pisces into zebrafish plasmids (*pminiTol2-10UAS-E1B*): 5′-CACGAATTCACAGCCACCGGTGCCACCAT GGCCCTCCAGAAGAAG-3′ (forward primer), and 5′-CTATAGTTC TAGAGGCTCGAGCTACTTAGCGCTGGTGTACTTGGTGATG-3′ (reverse primer). To create the *pminiTol2-10UAS-E1B-Pisces*, mCherry was replaced with the photoconvertible protein mMaple.

### Cell cultures and transfections

H1299 cells (TCHu160) and HEK293 cells (GNHu43) were validated to be mycoplasma contamination-free and obtained from the cell bank of the Chinese Academy of Sciences. H1299 cells and HEK293 cells were maintained in RPMI 1640 media (Gibco) and DMEM high glucose (Gibco) with 1% penicillin-streptomycin

(10,000 U/ml) and 10% FBS (ExCell Bio), respectively. Both cell lines were cultivated at 37 °C in a humidified atmosphere of 95% air and 5% $CO_2$.

For transient transfection of H1299 cell culture, the cells were plated on a transparent 96-well plate at ~70% confluency, and plasmid transfections were carried out using Lipofectamine 3000 (Invitrogen), following the manufacturer's instructions. To co-express pcDNA3-H2B-PhoCl and pcDNA3-NES-jRGECO1a in HEK293 cells, cells were seeded in a 35 mm glass-bottom 4-well plate (Cellvis) at ~50% confluency. Equal amounts of both plasmids (0.1 ng per well) were transfected into cells using Hieff TransTM Liposomal Transfection Reagent (YEASEN) according to the manufacturer's protocol.

### In situ PhoCl activation with a home-made LED setup
The schematic of the custom-made LED setup is shown in Supplementary Fig. 2b. A 40 W LED lamp (~405-nm) and other electronic devices were purchased from local suppliers. To test the translocation efficiency, the 96-well plate (24 h post-transfection) was wrapped with parafilm (Sigma) and placed in a container with sterile water for in situ activation with the custom LED setup. The stimulation cycle was 1.5-s on and 4-s off for 8 min, with a stimulation intensity of ~1.4 mW/mm². For 6 days-post-fermentation (dpf) zebrafish larvae, activation was performed in a 6 cm petri dish under continuous 2-min illumination from the LED lamp (~0.2 mW/mm² and ~1.4 mW/mm² for partial and complete activation, respectively).

### Cell culture imaging
To evaluate the light-inducible protein translocation system, high-content imaging was performed using the Operetta CLS (PerkinElmer) equipped with a 10× objective (N.A. 0.3). After photo-activation with the custom LED setup, cells in the 96-well plate were fixed with 4% PFA (Coolaber) for 2 h, followed by nuclear staining with 2 μg/ml DAPI (Sigma-Aldrich) in PBS buffer (Mxbioscience LLC) for 30 min. Cells were then washed three times with PBS buffer before imaging. Fluorescence signals of DAPI, PhoCl, and mCherry were detected using excitation filters 370/30, 475/30, and 545/30, and emission filters 465/70, 525/50, and 610/80, respectively. Image analysis was performed using the Operetta analysis program. The nuclear localization is defined as the nuclear signal relative to the whole cell signal. The translocation efficiency is determined by measuring the change in the mean ratio of the cytosolic signal to the total cellular signal in all cells, before and after activation.

Other cell culture imaging experiments were conducted using an inverted Nikon A1 confocal microscope with a PlanApo λ 40 ×objective (N.A. 0.95). Cells were plated on a 35 mm glass-bottomed dish (Cellvis), and the culture medium was replaced with HEPES buffer before experiments. For calcium imaging in HEK293 cells, 3 μM acetylcholine was added into cells after 1 min of steady state fluorescence recording. To detect Pisces0 performance in H1299 cells, the 405-nm laser (7.5 μW) was used for a 2-min continuous scanning during imaging. Fluorescence was excited using 488-nm and 561-nm lasers, and emission was detected at 500–550 nm and 570–620 nm, respectively, using a photo-multiplier tube (PMT) in a 1024 × 1024 format with 12-bit depth. All data were analyzed using ImageJ Fiji (1.53c).

### Zebrafish husbandry and preparation
Transgenic zebrafish was generated by the Tol2 transposase-based approach as reported previously[53]. In brief, 25 ng/μl of plasmid DNA and 25 ng/μl of Tol2 transposase mRNA (1 nL) were co-injected into zebrafish embryos at the one-cell stage using an air-puffed MPPI-3 pressure injector (ASI). Zebrafish larvae were kept in egg water under dark or ambient light environment (~250 lux) in the incubator for 3 or 6 days before experiments. Zebrafish embedding, tissue dissection, and cell sorting were conducted under ambient light conditions (~250 lux).

### Zebrafish lines
The study used the Nacre and AB wild-type strains and the transgenic reporter strains TgBAC(gng8:GAL4FF)^c426 abbreviated as Tg(gng8: Gal4FF)[54], Tg2(elavl3:GCaMP6s)^a13203 abbreviated as Tg(elavl3:GCaMP6s)[55], Tg(isl2b.2:Gal4-VP16,myl7:EGFP)^zc60 abbreviated as Tg(isl2b.2:Gal4-VP16, myl7:EGFP)[56], Ki(dbh:GAL4-VP16)^ion36d abbreviated as Ki(dbh:Gal4-VP16)[28], Tg(elavl3:GAL4-VP16)^ion8d abbreviated as Tg(elavl3:Gal4-VP16)[57], Tg(elavl3: NES-jRGECO1a)^ion72d abbreviated as Tg(elavl3:NES-jRGECO1a)[58]. The Tg(10xUAS-E1B:Pisces)^ion274d fish line, abbreviated as Tg(10UAS-E1B:Pisces), was developed by Rong-Kun Tao in Jiu-Lin Du's laboratory.

### In vivo imaging
For imaging neuronal morphology, zebrafish larvae were paralyzed in sterilized egg water (5 mM NaCl, 0.17 mM KCl, 0.33 mM $CaCl_2$, 0.33 mM $MgSO_4$, 1 mM pancuronium dibromide (Selleck)) and mounted dorsal side up into 1.5% low-melting agarose on a glass-bottom dish under a stereoscopic microscope. Imaging of the zebrafish larvae was performed using an Olympus FV3000 upright confocal microscope equipped with an XLUMPlanFLN 20× water-immersion objective (N.A. 1.0) and controlled via FLUOVIEW FV31S-SW software.

To compare the green and red fluorescence of PhoCl and Kaede in neurons, zebrafish larvae under different conditions (dark, ambient light environment, and LED illumination) were excited with 488 nm and 561 nm lasers. Z-stack images were captured with a field of 1024 × 1024 pixels using emission filters for 500–550 nm and 570–620 nm.

To test the impact of H2B-PhoCl on physiological calcium response in neurons, 6-dpf Tg(elavl3:NES-jRGECO1a) zebrafish larvae mosaically expressing neuronal H2B-PhoCl were subjected to a 5-trial electronic field stimulation (6 mA, 5 ms, 1 Hz, three pulses) with 60 s interval. The amplitude of jRGECO1a responses in PhoCl-positive neurons was compared with that of adjacent PhoCl-negative neurons.

To detect the functional identity of tectal neurons, Tg(elavl3:Gal4-VP16);Tg(UAS-E1B:Pisces);Tg(elavl3:GCaMP6s) zebrafish larvae were embedded with dorsal side up in 1.7% low-melting agarose without muscle paralysis. In vivo calcium imaging was performed on a cropped ROI from a 512×512-pixel image. Neuronal activities were imaged with 2 optical slices at 2–4 μm intervals at ~1 Hz for obtaining proper cytosol calcium activities. Visual stimulation was delivered by a red LED, with a 60-s dark period, followed by 10 trials of alternating 20-s light illumination and 20-s darkness. The LED was triggered by the AMPI Master-8 pulse stimulator in train mode and temporally aligned with calcium imaging. Calcium response was analyzed by manually outlining a cytosolic ROI around the nucleus of the tectal neuron for improved signal changes, since the nucleus constitutes a significant proportion of the soma region in the neurons of zebrafish larvae. Neuronal response subtypes were identified by averaging calcium responses across trials.

### Pisces activation, morphology tracing, and visualization of single neurons
For single-cell activation of Pisces, pre-activation z-stack images were captured, followed by previewing a neuron of interest (NOI) under a zoomed-in view at a spatial resolution of 0.2–0.3 μm/pixel. The NOI was selected by drawing a region of interest (ROI) of the neuron nucleus and then activating it using a 405-nm laser. Activation parameters were set to either 0.5 μW for 30–60 s with a 2 μs/pixel dwell time, or 1.5 μW for 10 s with a 10 μs/pixel dwell time. For enhanced activation precision, the process could be repeated 3–5 times with smaller ROIs (0.5–1 μm diameter) and shorter stimulation duration (5–15 s), ensuring complete activation of Pisces in the entire nucleus. Following activation, Z-stack images were acquired in 1 μm axial steps in a 1024 × 1024 format using bright-Z mode, capturing the complete morphology of the NOI in zebrafish larvae.

Tracing the morphology of ventral neurons or neurons within densely populated regions presents challenges due to the increased risk of off-target activation. To ensure specific and accurate labeling, precise neuronal activation protocols must be followed. This includes anesthetizing the fish to maintain a stable focal plane, using a low-intensity laser delivered in repeated short pulses, and carefully avoiding activation of target neurons located near other Pisces-positive cells directly above or below the focal plane. Despite occasional unintended activation of mMaple by scattered light, particularly in deeper tissues, the scattered light is insufficient to activate PhoCl, which minimizes the disturbance of nonspecific red nuclear mMaple signals and preserves the specificity of fiber labeling in the target neuron.

To acquire the morphology of functionally identified tectal neurons, calcium imaging was conducted simultaneously with red light stimulation, as described above. Neurons with specific light responses were selected, and their nuclear Pisces were activated, followed by full neuronal morphology imaging.

For subsequent fluorescence image processing, we employed a previously reported semiautomatic process using the FIQA2 code implemented in ImageJ Fiji[59]. This workflow included background subtraction, skin removal for improved clarity, generation of image stacks, and Z-projection images. To visualize neuronal morphology in detail, a 3D reconstruction was performed using Imaris software X64 9.0.1 (Bitplane, Belfast, United Kingdom).

### Morphology tracing by single-cell electroporation of Alexa Fluor 488

In sparsely labeled 5–7 dpf *Tg(elavl3:Gal4-VP16);Tg(UAS-E1B:Pisces)* larvae, one Pisces-positive neuron was photoactivated, followed by morphology imaging within 2–3 h. The larva was then paralyzed, followed by incisions on the skin. Alexa Fluor 488 dye (dissolved in DMSO to 1 mM stock and diluted to 200 μM working concentration with external solution) was delivered with a glass electrode (-1 μm tip opening) and electroporated into the activated neuron using Axon Instruments' Axoporator® 800A (parameter setting: Voltage: −10 mV; Train: 500 ms; Frequency: 100 Hz; Contrast: 40; Width: 1 ms; DC offset: 0.1 V). Morphological imaging was performed -1–2 h after electroporation. Neuronal morphology labeled by both Pisces and Alexa Fluor 488 was captured as described above.

### Morphology reconstruction, image registration, and analysis

Single-neuron morphologies were semi-automatically reconstructed using the Simple Neurite Tracer plugin in ImageJ Fiji, as previously described[60]. Tracing began at the soma and extended outward in high-resolution NRRD images centered on the cell body. The averaged morphology stack from each time series was aligned to a 6-dpf *Tg(HuC:H2B-GCaMP6f)* template brain (with annotated regions) using Symmetric Normalization (SyNRA) in Advanced Normalization Tools[60,61] (ANTsPyX v0.3.728) with default parameters on Ubuntu 20.04.5. To analyze layer-specific neurite distribution, we used the *Ki(isl2b:Gal4FF);UAS:tdTomato-CAAX* line to establish a layered optical tectum subregion based on retinal ganglion cell (RGC) projections.

Neuron morphologies and brain region data were exported in SWC and STL formats. Projection intensity was estimated by quantifying total fiber length within segmented regions and visualized with custom 3D software. A dual-channel reference template was generated from *elavl3:H2B-GCaMP6s;vglut2a:DsRed* larvae. Confocal stacks were resampled to isotropic resolution and registered via shape-based averaging and deformable registration using ANTs. Rigid and affine alignment was followed by iterative deformable warping, with convergence assessed via normalized cross-correlation (NCC) scores. The registration pipeline included two parallelized steps: rigid/affine registration, followed by deformable warping. Background fluorescence was excluded using a fish-body mask generated via the Computational Morphometry Toolkit (CMTK) and manually refined.

### Tissue dissociation, cell sorting, and SMART-Seq3

We used *Tg(elavl3:Gal4-VP16); Tg(UAS-E1B:Pisces); Tg(elavl3:GCaMP6s)* transgenic zebrafish line at 6-dpf for transcriptome profiling. Neurons were selectively activated in one of three brain regions: the left habenula, right habenula, or midbrain. Each larva had neurons activated in only one brain region. Activated larvae were anesthetized using 0.2% MS-222 (Sigma) and placed on a soft agar plate immersed in pre-cooled freshly prepared choline chloride solution (92 mM Choline chloride, 2.5 mM KCl, 1.2 mM NaH$_2$PO$_4$, 30 mM NaHCO$_3$, 20 mM HEPES, 25 mM Glucose, 2 mM Sodium ascorbate, 2 mM Thiourea, 3 mM Sodium pyruvate, 10 mM MgSO$_4$, 0.5 mM CaCl$_2$, 12 mM N-Acetyl-L-cysteine, mOsm 310, pH 7.4). The dorsal skull and eyes were carefully removed with forceps, followed by incisions were made caudal to the olfactory bulbs and rostral to the midbrain to isolate the brain tissue. The dissected brains were transferred into choline chloride solution containing 20 U/mL papain and 100 U/mL DNase I (Worthington) at 37 °C for 20 min, followed by gentle trituration with fire-polished glass Pasteur pipettes to create a single-cell suspension. To remove any remaining cell aggregates, the suspension was passed through a 40 μm cell strainer (Nunc).

Neurons were sorted using a BD FACS Aria Fusion Flow Cytometer (BD Biosciences) equipped with 488-nm and 561-nm excitation lasers. Cells were plotted for green (GCaMP6s and inactivated Pisces) and red (activated Pisces) fluorescence to gate the activated cell subpopulation. Single cells gated as high red signal and positive green signal were sorted into individual wells of 8-PCR strips containing 3 μl of lysis buffer (0.15%) Triton X-100 (Sigma), 5% PEG (Sigma), 0.5 U/μl recombinant RNase inhibitor (RRI) (Takara), 0.5 mM dNTP (Thermo Scientific), 1 μM Smart-seq3 oligo-dT primer (5′-biotin-ACGAGCATC AGCAGCATACGA T30VN-3′), using single-cell sort mode with a 100 μm nozzle[62]. Sorted single-cell samples were immediately spun down, frozen on dry ice, and stored until library preparation for sequencing.

The cDNA synthesis was conducted for 21 PCR cycles. Cell lysis, RNA denaturation, and reverse transcription were carried out for library preparation[62]. Libraries were subject to paired-end sequencing on the Illumina Novaseq xplus platform, yielding an average of approximately 3 GB of raw data per library.

### Multiplexed fluorescence in situ hybridization (easiFISH) using a customized hybridization chain reaction (HCR) protocol

To prepare zebrafish larvae for easiFISH, specimens were initially fixed in 4% paraformaldehyde (PFA) for 12 h, then transferred into PBST for storage[63]. Following initial fixation, a secondary 15-min post-fixation step in 4% PFA was conducted, followed by overnight dehydration using sequential incubations in 50% and 70% ethanol, followed by two rinses in 100% ethanol and a PBS wash. Subsequent permeabilization involved treatment with proteinase K (10 μg/mL) for 10 min at 37 °C, halted by two PBS washes. To initiate hybridization, specimens underwent a 10-min pre-incubation at 37 °C in hybridization buffer, preceding an overnight incubation exceeding 24 h at 37 °C within a probe solution comprising hybridization buffer and custom-designed Sangon Biotech probes targeting *gng8* (ID: 553609) and *tac3a* (ID: 100320280). Unbound probes were then removed using probe wash buffer at 37 °C and 5× SSCT at room temperature. For signal amplification, samples were initially equilibrated in amplification buffer, then subjected to overnight incubation in the dark at room temperature with snap-cooled, spectrally stable fluorescent hairpins (Molecular Instruments; two distinct sets custom-labeled with Alexa647 dye), culminating in washing away excess hairpins using 5× SSCT. Processed specimens were maintained in light-protected conditions at 4 °C until microscopic imaging.

### Transcriptome analysis

The preliminary data analysis was performed by Arenta Life Sciences. In brief, FastQC was used to examine the quality of the reads after

sequencing. Reads were aligned to Danio_rerio (GRCz10.91) using Hisat2 (v2.2.1) with the default parameters. StringTie (v1.3.3b) was used to quantify the exonic read counts to generate a cell-by-gene count matrix.

Data analysis was performed using Seurat (v4.3.0) on R (v4.1.3). Cells with more than 1000 genes were kept for analysis. Habenula cells expressing the gng8 gene were specifically kept. Gene counts were normalized and scaled by the "SCTransform" function. Principal component analysis (PCA) was conducted on scaled data for the top 2000 highly variable genes. Dimensional reduction and data visualization were performed using uniform manifold approximation and projection (UMAP) and t-distributed Stochastic Neighbor Embedding (t-SNE), which utilized principal components that accounted for more than 85% of the variance. Seurat's anchor-based integration approach was used using FindIntegrationAnchors with 2000 features to integrate private and published habenula datasets[38]. The "FindAllMarkers" function was used to identify differential gene expression across various regions. The "compareCluster" function of clusterProfiler (v4.2.2) was used to enrich the functions of differentially expressed genes.

### Statistics & reproducibility

For analysis of calcium imaging data, $\Delta F/F_0$ of different neuron types were calculated as described above. $\Delta F/F_0$ traces represent the average neuronal activity across all trials in a given session. The normality of data was first examined with the Kolmogorov–Smirnov test or Sharpiro–Wilk test. Chi-square and unpaired Student's $t$-test were used for unpaired group comparisons in (Figs. 1c, 5d, h) and in (Supplementary Figs. 1j and 2i), respectively. A two-way ANOVA was used for paired comparisons (Supplementary Fig. 1f). A paired Student's $t$-test was used for group comparisons (Supplementary Fig. 5d, e). No statistical method was used to predetermine sample size. For all experiments, the replicates are biological from independent experimental units or subjects. For all data, error bars or shaded regions indicate the SEM, and detailed $P$-values are provided in the figures (unless the value is smaller than 0.0001). All statistical analyses and graphical representations were performed using GraphPad Prim (V9.5.1) or MATLAB R2018a (Mathworks). The figures were compiled and finalized using Adobe Illustrator 2020.

### Reporting summary

Further information on research design is available in the Nature Portfolio Reporting Summary linked to this article.

## Data availability

The single-cell transcriptome datasets have been deposited in the National Genomics Data Center under accession number PRJCA031958. The whole habenula dataset used in this study can be accessed on the Gene Expression Omnibus (GEO) under accession code GSE105115. Plasmid pminiTol2-UAS-E1B-Pisces is available on the WeKwikGene plasmid repository at Westlake Laboratory, China (https://wekwikgene.wllsb.edu.cn/). Zebrafish lines and all other plasmids generated in this study are available upon request. The zebrafish line generated in this study is preserved at the Center for Excellence in Brain Science and Intelligence Technology (CEBSIT) and is available from the corresponding authors upon request and signature of a Material Transfer Agreement between the CEBSIT and the requestor. Due to file sizes, raw imaging data are also available on request, and requests will be processed and fulfilled within four weeks. Source data are provided with this paper.

## Code availability

The custom code used in this study, FIQA2, has been previously published[59] and is accessible at https://github.com/KeJiiii/FIQA/releases. The custom 3D visualization software used in this study was developed using the ParaView framework and is currently under active

development in our laboratory. For collaborative or academic use prior to publication, please contact the lead corresponding author directly with a detailed request.

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

## Acknowledgements

We thank Drs. Fu-Ning Li, Chun-Feng Shang, Han-Yang Hu, Wei-Xi Feng, Hua-Xing Zi, Yu Mu, and Shuo Chen for their kind support. We appreciate the technical assistance from Tian-Lun Chen, Kui Wang, Qi-Meng Zhao, Hao Deng, Hong-Li Wan, Hui Zhang, Meng-Meng Jin, Wen-Hui Xu, Chao Li, and Qian Hu. We acknowledge Dr. Xu-Fei Du for her assistance with neuronal morphology registration on the brain template. We thank the Facility of Mapping Brain-wide Mesoscale Connectome, Single-Cell Typing Platform, Molecular and Cellular Biology Core Facility, and Optical Imaging Facility at CEBSIT (Institute of Neuroscience) for technical support. This work was supported by the National Science and Technology Innovation 2030 Major Program of the Ministry of Science and Technology (2021ZD0202203 to R.T., 2021ZD0204500 and 2021ZD0204502 to J.D.), National Natural Science Foundation of China (32171090 to R.T., 32321003 to J.D.), Shanghai Science and Technology Commission (21ZR1482600 to R.T.), 2023 Youth Innovation Promotion Association CAS (to R.T.), and Shanghai Municipal Science and Technology Major Project (18JC1410100 to J.D.).

## Author contributions

Conceptualization: R.T., J.D.; Design and validation of Pisces: R.T.; Methodology: R.T., L.S., Q.Y., Y.H., Y.C., C.G.; Investigation: R.T., L.S.; Visualization: R.T., L.S., Q.Y., Y.H., Y.C.; Funding acquisition: R.T., J.D., M.W.; Project administration: R.T., J.D.; Supervision: R.T., J.D., Y.S.; Writing - original draft: R.T.; Writing - review & editing: R.T., J.D., L.S., Q.Y., Y.H., Y.S., Y.C.

## Competing interests

R.T., J.D., L.S., and Q.Y. have submitted a patent application to the Chinese patent office pertaining to the development and application aspect of Pisces in this work (application number: CN202510316909.6). The remaining authors declare no competing interests.
