## [Transparent Peer Review file · Nature Communications]

Designed optogenetic tool for bridging single-neuronal multimodal information in intact animals

Corresponding Author: Professor Jiu-Lin Du

Version 0:

Reviewer comments:

Reviewer #1

(Remarks to the Author)

Tao, Du, and coworkers, report the development of a new technique that they designate as “photo-inducible single-cell labeling system” or Pisces. This technique enables the labeling of specific neurons in vivo, as demonstrated in zebrafish larvae. Once a neuron has been labeled using Pisces, fluorescence imaging can be used to determine the entire morphology of the neuron. To develop Pisces, the authors first optimized a PhoCl-mCherry fusion construct that translated from the nucleus to the cytosol upon 405 nm illumination. After optimizing the construct with mCherry, the mCherry gene was replaced with the mMaple gene to create the final Pisces construct.

Pisces is a fusion of two proteins, mMaple and PhoCl, both of which can convert from green to red fluorescence. In the case of the mMaple fusion partner, the red fluorescence is stable following photoconversion. In the case of the PhoCl protein, the red fluorescence is transient as the protein will dissociate into two fragments, releasing the NLS sequence that initially kept the Pisces protein localized to the nucleus. Upon release of the NLS, the red form of mMaple can leave the nucleus and freely diffuse throughout the neuron, enabling tracing of the entire morphology of the cell using fluorescence imaging.

Although I am familiar with PCFPs and related technologies, it took me a while to really understand the design and advantages of Pisces relative to other technologies. As I understand it, the main advantage is that all of the PCFP is initially confined to the nucleus and therefore can be photo converted using laser activation at a singular spot. In so doing, the green protein is converted to red, and simultaneously the NLS is released from the protein. This protein can then be transported out of the nucleus and diffuse throughout the volume of the neuron. In contrast, a standard PCFP confined to the nucleus could be photoconverted to a red fluorescent species to mark the nucleus, but it would not fill the cytosol and reveal the overall cell morphology. Furthermore, a freely diffusing PCFP could be photoconverted in the soma to reveal the overall cell morphology, but this would require more illumination spread over a longer period of time as the experimenter waits for all of the protein to diffuse through the illumination spot. As commented below, I believe that the authors could make the manuscript more accessible to non-experts by doing a better job of explaining and illustrating the features and advantages of Pisces.

The authors provide very compelling examples of using Pisces for neuronal labeling in larval zebrafish, including labeling in the optic tectum, the habenula, and the locus coeruleus. Furthermore, the authors demonstrated that labeling remained stable over many hours and that multiple neurons can be labeled sequentially, even when they are adjacent to each other. To combine Pisces labeling with functional characterization, the authors co-expressed Pisces with cytosolic GCaMP6s. With these co-expressed genes, the authors were able to characterize the activity of specific neurons in the optic tectum, classifying them in one of five categories, and then determine the complete morphology of these characterized neurons. Finally, the authors were able to use FACS to isolate individual Pisces-labeled neurons from the dissociated brain tissue of larval zebrafish, and analyze the transcriptome use single cell RNA sequencing.

Overall, it is my opinion that Pisces represents an interesting and useful new addition to the toolbox for photo-induced cell labeling. The main strengths of this manuscript are the uniqueness of the Pisces design and the multiple in vivo demonstrations of Pisces that were performed. Of the demonstrations, the single cell RNA sequencing of neurons that had been characterized both in terms of function and morphology is particularly impressive. The manuscript is generally well-written, the figures are clear, and all of the work appears to have been performed with the highest level of technical expertise.

It is my opinion that this manuscript will be publishable once the following points have been addressed:

Major concerns:

1. The authors provide data to show that an advantage of PhoCl relative to Kaede is that Kaede is partially activated under ambient lighting, while PhoCl is not. On lines 77-82, some additional favorable properties of PhoCl are stated. Specifically, the authors demonstrate that PhoCl is not activated with wavelengths from 445-640 nm and does not interfere with calcium imaging. As the paragraph is currently written, it is implied that these are advantages relative to Kaede. Were the experiments in Figure 1d,e and Extended Data Figure 1a-j also performed with Kaede? If so, this data should be shown. If they were not performed, the authors should make sure this is clear in the main text and that Kaede may not necessarily perform worse than PhoCl in these experiments.
2. As described in comment 1, the authors initially demonstrated some advantages of PhoCl relative to Kaede. However, they ultimately ended up using a different PCFP (mMaple), in the final version of Pisces. Unfortunately, the authors did not provide data to demonstrate that mMaple share the advantages of PhoCl. For example, is mMaple activated at 445-640 nm? Is mMaple compatible with Ca²⁺ imaging? Generally speaking, the rationale for choosing mMaple (relative to other available PCFPs) could be better explained.

Minor concerns:

3. I assume that 'hinging' in the title is a typo? I struggle to make sense of the intended meaning and a different word choice may be appropriate.
4. I understand that there is a bit of flexibility in the definition of "optogenetic tool", which is most commonly defined only as a tool for light-activated "control" of biology. Less commonly, it is sometimes defined as a tool for either "control or analysis". In my opinion, Pisces does not fit the most commonly accepted definition of an optogenetic tool as it does not enable control of the cell biology. I would suggest a different word choice in the title and in the main text to prevent confusion among readers.
5. I suggest that the authors could better explain the rationale for the Pisces design in the introduction, and more clearly state the advantages/disadvantages in the Discussion section. Along these lines, I suggest that it would be very helpful for readers if the authors prepared a schematic figure to demonstrate the advantages of Pisces relative to other technologies. Specifically, I think the figure should schematically represent the expected outcomes with a PCFP confined to the nucleus (with an NLS), a PCFP excluded from the nucleus (with an NES), a PCFP exchanging between the cytoplasm and the nucleus (with no tag), Pisces0.1, and the final Pisces construct. I feel like such a figure could help to make the article much more accessible to experts and non-experts alike.
6. One of the stated disadvantages of PCFPs is the slow diffusion from the soma to the neurites (lines 47-49). The authors demonstrate that Pisces exhibits sufficiently fast neurite labeling (lines 118-121). Is this any different than what would be expected for soma-localized photoconversion of a freely diffusing PCFP?
7. On lines 134-136, the authors state that the rapid diffusion is facilitated by the NES. The precise meaning of this statement is not clear to me, and it is also not clear if there is any evidence to support it. Are the authors suggesting that the diffusion is more rapid with an NES than without an NES? If so, is there a reasonable mechanistic explanation?
8. To discover Pisces0.1, the authors seem to have tested more than 30 different constructs (as listed in Extended Data Fig. 2c) to find the best combination. The exact number of constructs tested should probably be mentioned in the main text.
9. Confocal is misspelled in all instances where it occurs inside a figure.

Reviewer #2

(Remarks to the Author)

Uncovering neuronal cell types through their morphology, cell body location, function, and transcription is a major focus in neuroscience and developmental biology. Therefore, establishing a method to bridge detailed cellular structures, neural function, and transcription at single-cell resolution is both important and challenging. Previous studies utilizing photoactivated GFP (paGFP) allowed for the visualization of morphology at the single-cell level following functional imaging with GCaMP. However, due to the similar emission wavelengths of GCaMP and paGFP, it was difficult to separate activated and non-activated neurons. Additionally, lack of precision in targeting the activation wavelength posed an issue. In the present study, the authors engineered a light-inducible color-changing molecule called Pisces, which does not overlap with GCaMP's excitation properties and is less prone to leakage due to its self-cleavage site. By combining these two features, they successfully labeled neurons within a short time window, which is critical for labeling cells in a developing organism. They demonstrated that multiple neurons could be labeled sequentially, thereby nicely reconstructing neural structures.

Although the authors successfully demonstrated the ability to connect morphology and neural function at the single-cell level, the relationship between morphology and transcription is not presented in the same context; instead, only the transcriptional cell types within groups of neurons are shown. The authors provided a structural analysis of selected neurons

in the habenula in Figure 3, and a functional and morphological analysis of selected neurons in the midbrain in Figure 5. In Figure 6, they labeled habenula and midbrain neurons broadly and sparsely in different fish, followed by single-cell RNA sequencing (RNA-seq). Consequently, we can only very roughly deduce the correspondence of cell types in the habenula, but not with the same precision in the midbrain. Notably, the resolution of cell types identified through functional and transcriptional analyses was quite different.

While it is understandable that there are technical difficulties in matching single-cell RNA-seq to the morphotypes of neurons, the authors claim to integrate morphology and molecular information for individual neurons (as illustrated in Figure 6i). Therefore, it is necessary to demonstrate the correspondence between morphology and transcription. One suggestion is first classifying the neurons by morphology and transcription independently, and investigating if the two classifiers match or not.

Additionally, the following are some other points needed to be described or considered in the manuscript,

(1) Spatial limitation of visualization:

The manuscript only presents data on neurons located in the dorsal part of the brain. This raises concerns regarding the completeness of the analysis, as the ventral part is not represented. Please clarify if there are inherent limitations in the visualization technique that preclude the examination of neurons in this region. A more comprehensive exploration of both dorsal and ventral neurons would enhance the study's impact.

(2) Homogeneity of expression:

The expression levels of the *Pisces* protein across neuronal populations appear to be homogeneous, yet the current data lacks a quantifiable measure. Since the fish examined are first-generation injected fish, it is plausible that expression varies among neurons. I recommend providing a quantification of the percentage of neurons expressing the plasmid to substantiate claims regarding the technique's efficiency and robustness.

(3) Registration quality:

The quality of registration for imaging stacks is crucial for accurately interpreting neuronal structure and connectivity. The methods section does not adequately describe the registration approach employed in this study. Given the incomplete expression of *Pisces* in the larvae depicted in Figures 1, 2b, 3a, and 4, I encourage the authors to provide more details on the registration process and to ensure that the positional accuracy of cell bodies and projections is addressed explicitly.

(4) Development of the larval brain during labeling:

The manuscript should consider discussing the developmental changes occurring in the larval brain over the labeling period provided. It is important to address potential effects of cell division or the alteration of neural projections that could influence results at different ages. In addition, whether transcription changes before and after the labeling is critical to be mentioned. Clarifying the applicability of the *Pisces* technique across various temporal scales would greatly enhance the robustness of the findings.

Minor corrections in figures:

(1) Figure 3a: In the leftmost schematic drawing of the fish larvae, the location of the green blobs appears to correspond to the olfactory bulb rather than the habenula, which is misleading. This issue is also present in Extended Data Fig. 3.

(2) Figure 2: Figure 2 contains two instances of "b" and no instance of "c."

(3) Figure 5f: I observe a few projections that seem to cross between the two hemispheres, yet they are labeled as the ipsilateral projection dataset. Could you please clarify this?

Questions in the videos:

Video1: Since the ventral part of the brain does not express signals, it is not convincing that the cellular morphology of neurons in the whole brain is trackable or not. If there is a spatial limitation in the technique, then it should be mentioned in the manuscript. It would be great if an overlay of the schematic drawing of the whole brain was shown with the imaging stacks.

Video2: The manuscript said the single neurons are labeled, but I can see two blobs, which should be the cell bodies, in each hemisphere. Is this an artifact of imaging or two cells are visualized in each hemisphere? And if it's the latter, then it's hard to say this is a single-cell projection.

Video3: It would be more convincing if the signals are overlaid with the skeletons of the tracked projections.

Reviewer #3

(Remarks to the Author)

Rong-Kun and co-authors presents a genuinely elegant manuscript describing a new approach to photo-label single neurons in-vivo, a method they termed Pisces. The methods allows to acquire functional, morphological and transcriptomic data on a single neuron level. I am certain that the method has the potential to dramatically push forward neuronal-circuits studies.

I would like to point to two fundamental issues that require in-depth revision of the current manuscript.

First and most important to me, is the fact that the authors claim that the labeling is "complete" neurons (e.g. [... capability of Pisces in outlining the complete morphology...This demonstrates Pisces' fidelity in tracing entire neuronal morphologies... pp5 lines 122-9] OR [... we demonstrate that Pisces enables high-resolution and clear tracing of the complete morphology of multiple neurons... , pp10, line 265-6]. The data provided to support this claim (e.g Extended Data Fig 3) is not sufficient for this reviewer to appreciate the completeness of the photo-activated labeling. Such a bold statement has to be accompanied with a clear comparison with labeling the same neurons with an independent technique of their choice.

Second, the authors demonstrate the ability to perform this approach in zebra-fish, they suggest the method will be "compatible with other small animal models" albeit no support for such a claim is provided. Given the extensive use of rodents in Neuroscience research, it would be a high priority to provide evidence that this approach is indeed compatible in mammals.

I hope the above comment can be swiftly addressed.

Reviewer #4

(Remarks to the Author)

Version 1:

Reviewer comments:

Reviewer #1

(Remarks to the Author)

The authors have addressed all of the reviewer comments to my satisfaction. I recommend that the manuscript be accepted in its current form.

Reviewer #2

(Remarks to the Author)

The authors have thoroughly addressed all of our previous comments with a comprehensive set of new experiments and analyses. Their revisions have significantly strengthened the manuscript, and we appreciate the careful and thoughtful response to each point raised. We are satisfied with the improvements and have no further concerns. We support the publication of this work in its current form.

Reviewer #3

(Remarks to the Author)

The authors have thoroughly addresses my comments and I have no further input to provide to this version.

Reviewer #4

(Remarks to the Author)

Responses to the Reviewers

(Reviewers' comments are in black and responses are in blue)

Reviewer #1 (Remarks to the Author):

Tao, Du, and coworkers, report the development of a new technique that they designate as “photo-inducible single-cell labeling system” or Pisces. This technique enables the labeling of specific neurons in vivo, as demonstrated in zebrafish larvae. Once a neuron has been labeled using Pisces, fluorescence imaging can be used to determine the entire morphology of the neuron. To develop Pisces, the authors first optimized a PhoCl-mCherry fusion construct that translated from the nucleus to the cytosol upon 405 nm illumination. After optimizing the construct with mCherry, the mCherry gene was replaced with the mMaple gene to create the final Pisces construct.

Pisces is a fusion of two proteins, mMaple and PhoCl, both of which can convert from green to red fluorescence. In the case of the mMaple fusion partner, the red fluorescence is stable following photoconversion. In the case of the PhoCl protein, the red fluorescence is transient as the protein will dissociate into two fragments, releasing the NLS sequence that initially kept the Pisces protein localized to the nucleus. Upon release of the NLS, the red form of mMaple can leave the nucleus and freely diffuse throughout the neuron, enabling tracing of the entire morphology of the cell using fluorescence imaging.

Although I am familiar with PCFPs and related technologies, it took me a while to really understand the design and advantages of Pisces relative to other technologies. As I understand it, the main advantage is that all of the PCFP is initially confined to the nucleus and therefore can be photo converted using laser activation at a singular spot. In so doing, the green protein is converted to red, and simultaneously the NLS is released from the protein. This protein can then be transported out of the nucleus and diffuse throughout the volume of the neuron. In contrast, a standard PCFP confined to the nucleus could be photoconverted to a red fluorescent species to mark the nucleus, but it would not fill the cytosol and reveal the overall cell morphology. Furthermore, a freely diffusing PCFP could be photoconverted in the soma to reveal the overall cell morphology, but this would require more illumination spread over a longer period of time as the experimenter waits for all of the protein to diffuse through the illumination spot. As commented below, I believe that the authors could make the manuscript more accessible to

non-experts by doing a better job of explaining and illustrating the features and advantages of Pisces.

The authors provide very compelling examples of using Pisces for neuronal labeling in larval zebrafish, including labeling in the optic tectum, the habenula, and the locus coeruleus. Furthermore, the authors demonstrated that labeling remained stable over many hours and that multiple neurons can be labeled sequentially, even when they are adjacent to each other. To combine Pisces labeling with functional characterization, the authors co-expressed Pisces with cytosolic GCaMP6s. With these co-expressed genes, the authors were able to characterize the activity of specific neurons in the optic tectum, classifying them in one of five categories, and then determine the complete morphology of these characterized neurons. Finally, the authors were able to use FACS to isolate individual Pisces-labeled neurons from the dissociated brain tissue of larval zebrafish, and analyze the transcriptome use single cell RNA sequencing.

Overall, it is my opinion that Pisces represents an interesting and useful new addition to the toolbox for photo-induced cell labeling. The main strengths of this manuscript are the uniqueness of the Pisces design and the multiple in vivo demonstrations of Pisces that were performed. Of the demonstrations, the single cell RNA sequencing of neurons that had been characterized both in terms of function and morphology is particularly impressive. The manuscript is generally well-written, the figures are clear, and all of the work appears to have been performed with the highest level of technical expertise.

We sincerely appreciate the reviewer's thorough evaluation and positive feedback on our manuscript. We are especially grateful for the recognition of Pisces as a valuable tool for photo-inducible cell labeling and for highlighting the strength of our in vivo demonstrations, including single-cell RNA sequencing of functionally characterized neurons.

We acknowledge the reviewer's suggestion to improve the accessibility of our manuscript for non-experts. To address this, we have revised the introduction and figure legends to provide clearer explanations of Pisces' design, advantages, and applications. Additionally, we have refined our schematic illustrations to better communicate the key principles behind Pisces. We believe these modifications will enhance the manuscript's readability and make it more approachable for a broader audience.

It is my opinion that this manuscript will be publishable once the following points have been addressed:

Major concerns:

1. The authors provide data to show that an advantage of PhoCI relative to Kaede is that Kaede is partially activated under ambient lighting, while PhoCI is not. On lines 77-82, some additional favorable properties of PhoCI are stated. Specifically, the authors demonstrate that PhoCI is not activated with wavelengths from 445-640 nm and does not interfere with calcium imaging. As the paragraph is currently written, it is implied that these are advantages relative to Kaede. Were the experiments in Figure 1d,e and Extended Data Figure 1a-j also performed with Kaede? If so, this data should be shown. If they were not performed, the authors should make sure this is clear in the main text and that Kaede may not necessarily perform worse than PhoCI in these experiments.

We acknowledged the reviewer's concern regarding the implications of our statements on lines 77-82. To clarify, the primary demonstrated advantage of PhoCI over Kaede in our study is its reduced sensitivity to ambient light. The additional properties mentioned are only tested on PhoCI, such as its non-activation at 445-640 nm and lack of interference with calcium imaging, which have not been explicitly tested for Kaede in our experiments.

We have explicitly stated in the main text (lines 89-90) as follows.

We did not test the properties of nuclear Kaede in these experiments.

We believe that this revision will ensure the comparison between PhoCI and Kaede focuses solely on their sensitivity to ambient light.

2. As described in comment 1, the authors initially demonstrated some advantages of PhoCI relative to Kaede. However, they ultimately ended up using a different PCFP (mMaple), in the final version of Pisces. Unfortunately, the authors did not provide data to demonstrate that mMaple share the advantages of PhoCI. For example, is mMaple activated at 445-640 nm? Is mMaple compatible with Ca²⁺ imaging? Generally speaking, the rationale for choosing mMaple (relative to other available PCFPs) could be better explained.

We appreciate the request for a more thorough rationale behind selecting mMaple as the photoconvertible fluorescent protein (PCFP) in Pisces. Below is our clarified justification.

2.1 For example, is mMaple activated at 445-640 nm?

The optical properties of mMaple have been extensively characterized in the literature (McEvoy et al., 2012). The mMaple is a green-to-red photoconvertible fluorescent protein that undergoes efficient and irreversible photoconversion upon exposure to violet light (e.g., 405 nm), similar to PhoCl. In addition, like other photoconvertible fluorescent proteins (pcFPs), mMaple can also undergo primed conversion when illuminated simultaneously with 458 nm and 730 nm light (Turkowyd et al., 2017).

To our knowledge, there have been no reports of mMaple or other pcFPs being activated solely within the 445-640 nm range under standard imaging conditions. Therefore, we do not expect mMaple to be activated within this spectral window.

Interestingly, mMaple also exhibits reversible photo-switching behavior, similar to mClavGR2. Specifically, in its red fluorescent state, mMaple can be switched into a non-fluorescent dark state by green light illumination (e.g., 532 nm), and subsequently reactivated by exposure to 405 nm, 460 nm, or broad-spectrum white light (McEvoy et al., 2012).

Reference

- McEvoy, A. L. *et al.* mMaple: A Photoconvertible Fluorescent Protein for Use in Multiple Imaging Modalities. *Plos One* **7**, e51314 (2012).
- Turkowyd, B. *et al.* A General Mechanism of Photoconversion of Green-to-Red Fluorescent Proteins Based on Blue and Infrared Light Reduces Phototoxicity in Live-Cell Single-Molecule Imaging. *Angew. Chem. Int. Ed Engl.* **56**, 11634–11639 (2017).

2.2 Is mMaple compatible with Ca²⁺ imaging?

We recognize the importance of verifying that mMaple does not interfere with calcium imaging. Although we have not directly assessed the interaction between mMaple and calcium imaging in this study, several lines of evidence support their compatibility:

1. **Sequence and structural similarity:** PhoCl is a circularly permuted variant of mMaple, sharing high sequence identity and a nearly identical crystal structure (Zhang et al., 2017). Since PhoCl has been empirically shown not to interfere with calcium imaging (Extended Data Fig. 1c-j), it is reasonable to infer that mMaple is also non-disruptive.

2. **Empirical precedent:** Numerous studies have demonstrated the successful co-expression of GCaMP with other photoactivatable fluorescent proteins or RFP (e.g., PAGFP, PACherry, mCherry), without any reported interference (Cohen et al., 2022; Kramer et al., 2019; Lee et al., 2019). Given the conserved β -barrel structure and spectral properties shared among these proteins, including mMaple and Kaede, it is unlikely that mMaple would disrupt Ca^{2+} imaging.
3. **Spectral overlap and localization strategies:** While mMaple's emission spectrum partially overlaps with GCaMP6s, this limitation can be mitigated through spatial separation of signals (Fig. 4a,b). As demonstrated in this work, nuclear-localized mMaple (or PhoCl/Pisces) can be co-imaged with cytosolic GCaMP6s or red-shifted calcium indicators such as jRGECO1a, avoiding spectral cross-talk (Extended Data Fig. 1c-f).

Taken together, the available evidence suggests that mMaple is compatible with Ca^{2+} imaging under appropriate experimental settings.

Reference

- Zhang, W. *et al.* Optogenetic control with a photocleavable protein, PhoCl. *Nat. Methods* **14**, 391–394 (2017).
- Kramer, A., Wu, Y., Baier, H. & Kubo, F. Neuronal Architecture of a Visual Center that Processes Optic Flow. *Neuron* **103**, 118-132.e7 (2019).
- Lee, D., Kume, M. & Holy, T. E. Sensory coding mechanisms revealed by optical tagging of physiologically defined neuronal types. *Science* **366**, 1384–1389 (2019).
- Cohen, R. *et al.* A genetically targeted sensor reveals spatial and temporal dynamics of acrosomal calcium and sperm acrosome exocytosis. *J. Biol. Chem.* **298**, 101868 (2022).

2.3 Generally speaking, the rationale for choosing mMaple (relative to other available PCFPs) could be better explained.

We appreciate the reviewer's insightful suggestion. According to previous studies (McEvoy et al., 2012), mMaple is a high-performance, monomeric, green-to-red photoconvertible fluorescent protein optimized for multimodal imaging applications. Compared with other commonly used PCFPs, such as mEos2 and mClavGR2, mMaple exhibits several notable advantages, including superior photostability, enhanced folding efficiency, and excellent performance in super-resolution microscopy. These key properties are summarized in Table R1.

Feature	mMaple vs. Others
Photostability	Much higher green-state stability (>14-fold) and comparable red-state stability
Folding Efficiency	Higher in vivo folding and solubility
Brightness	Brighter in bacterial cells and similar in mammalian cells
Photoconversion Contrast	Higher red/green contrast after 405 nm (~2-4 fold)
Super-resolution Utility	Higher protein localization counts in PALM/STORM
Expression in E. coli	More efficient expression and maturation

Table R1. Summary of performance characteristics of mMaple in comparison with other PCFPs, based on prior reports (McEvoy et al., 2012).

Given these advantages, mMaple is particularly well suited for applications requiring robust expression, efficient photoconversion, and high-resolution imaging. Therefore, it was selected as an ideal candidate to replace mCherry for constructing an efficient morphological tracing tool, Pisces.

We have revised the corresponding statements in the Results section to clarify the rationale behind our choice of mMaple as follows (lines 116-121).

To optimize the system, we created an improved version called Pisces by replacing mCherry with the PCFP mMaple (Fig. 2c and Supplementary Note 1), which is a high-performance, monomeric, green-to-red photoconvertible fluorescent protein optimized for multimodal imaging applications and demonstrated superior photostability, enhanced folding efficiency, and excellent performance in super-resolution microscopy when compared with other commonly used PCFPs (such as mEos2 and mClavGR2).

Reference

McEvoy, A. L. et al. mMaple: A Photoconvertible Fluorescent Protein for Use in Multiple Imaging Modalities. *Plos One* 7, e51314 (2012).

Minor concerns:

3. I assume that 'hinging' in the title is a typo? I struggle to make sense of the intended meaning and a different word choice may be appropriate.

Thank you for raising this important point. We agree that "hinging" could imply an unintended mechanistic or structural dependency. "Bridging" might better reflect the pairwise associations demonstrated in the results.

We have revised the title as below:

Designed optogenetic tool for bridging single-neuronal multimodal information in intact animals.

4. I understand that there is a bit of flexibility in the definition of "optogenetic tool", which is most commonly defined only as a tool for light-activated "control" of biology. Less commonly, it is sometimes defined as a tool for either "control or analysis". In my opinion, Pisces does not fit the most commonly accepted definition of an optogenetic tool as it does not enable control of the cell biology. I would suggest a different word choice in the title and in the main text to prevent confusion among readers.

Thank you for this thoughtful critique. We agree that clarifying terminology is critical. The term "optogenetics" broadly refers to techniques using genetically encoded, light-responsive proteins to achieve spatial and temporal control of cellular physiology (e.g., ion channels, pumps, or proteins with light-dependent activity switches). While the majority of optogenetic tools directly regulate cellular activity (e.g., ion flux or signaling pathways), the field has expanded to include tools for light-controlled protein activity, such as PhoCl, which is categorized as a fourth class of optogenetic tools (alongside microbial opsins, light-regulated enzymes, and oligomerization systems)(Zhang et al., 2017).

Pisces, derived from PhoCl, enables light-dependent control of protein translocation between the nucleus and cytoplasm. While it does not modulate ion fluxes or signaling in the traditional sense, its mechanism aligns with the broader optogenetic paradigm of using light to manipulate protein behavior. Thus, we keep the terminology of "optogenetic" in the title.

However, to align with the reviewer's concern and avoid ambiguity for readers expecting "control" to imply direct functional modulation (e.g., neuronal firing or biochemical signaling), we have revised the terminology in the main text. Specifically, we now describe Pisces as a "light-controllable protein translocation tool" (lines 94 and 290), emphasizing its precise spatial regulation of protein localization rather than functional control.

We believe that this adjustment ensures clarity while maintaining consistency with the original definition of optogenetics and the reported classification of PhoCI-based systems.

Reference

Zhang, W. *et al.* Optogenetic control with a photocleavable protein, PhoCI. *Nat. Methods* **14**, 391–394 (2017).

5. I suggest that the authors could better explain the rationale for the Pisces design in the introduction, and more clearly state the advantages/disadvantages in the Discussion section. Along these lines, I suggest that it would be very helpful for readers if the authors prepared a schematic figure to demonstrate the advantages of Pisces relative to other technologies. Specifically, I think the figure should schematically represent the expected outcomes with a PCFP confined to the nucleus (with an NLS), a PCFP excluded from the nucleus (with an NES), a PCFP exchanging between the cytoplasm and the nucleus (with no tag), Pisces0.1, and the final Pisces construct. I feel like such a figure could help to make the article much more accessible to experts and non-experts alike.

We appreciate this valuable suggestion. In response, we have expanded the Introduction to provide a clearer rationale for the design of Pisces, highlighting the limitations of existing photoconvertible proteins and how Pisces addresses these gaps. We have also revised the Results and Discussion section to explicitly outline the advantages and limitations of Pisces in comparison with conventional PCFPs.

Additionally, as recommended, we have created a new schematic figure (Fig. R1.1, Extended Data Fig. 2j) that visually compares the expected localization and photoconversion outcomes for a PCFP with an NLS, NES, without any targeting sequence, Pisces0.1 and the final Pisces construct.

Fig. R1.1. Schematic representation of constructs (left) and expected outcomes (right) for photo-convertible fluorescent proteins (PCFPs) with different signal peptides and Pisces. (a) Nucleus-localized PCFP, (b) Nucleus-excluded PCFP, (c) PCFP without a signal peptide, (d) Pisces0.1, (e) Pisces. The purple triangle represents the 405 nm laser.

The revisions are as follows.

Introduction section (lines 62-67)

In this study, we designed a Photo-inducible single-cell labeling system (Pisces) by utilizing a nuclear chimeric protein composed of a photo-cleavable protein (PhoCl), a photo-convertible fluorescent protein (mMaple), and a balanced combination of nuclear localization signal (NLS) and nuclear export signal (NES). Upon PhoCl activation, the cleaved, photoconverted mMaple is actively transported by NES, enabling rapidly tracing the entire complex morphology of any single neurons in intact larval zebrafish.

Results section (lines 125-128)

*To clarify the working principle, we generated a schematic figure (**Extended Data Fig. 2j**) that compares the expected localization and photoconversion outcomes for a PCFP with an NLS, NES, without any targeting sequence, Pisces0.1, and the final Pisces construct.*

Discussion section (lines 290-295)

*Unlike traditional genetically encoded methods (see **Extended Data Figure 2j**), which exhibit limited effectiveness in ambient light conditions, require repeated activation in the soma region, and take days to weeks for complete labeling due to slow protein diffusion rate ($\sim 0.02 - 0.09 \mu\text{m/s}$), Pisces achieves precise manipulation by nuclear localization and rapid active, energy-dependent nuclear export by NES ($\sim 1 \mu\text{m/s}$).*

We believe these revisions significantly improve the accessibility of our method for both expert and non-expert readers.

6. One of the stated disadvantages of PCFPs is the slow diffusion from the soma to the neurites (lines 47-49). The authors demonstrate that Pisces exhibits sufficiently fast neurite labeling (lines 118-121). Is this any different than what would be expected for soma-localized photoconversion of a freely diffusing PCFP?

Thank you for this insightful comment. The distinction lies in the subcellular localization strategy and mechanism of protein redistribution between conventional PCFPs and Pisces.

1. The limitation of freely diffusing PCFP

In neurons, the soma predominantly comprises the nucleus, which occupies most of its volume. When freely diffusing PCFP is expressed throughout the neuron (soma and all processes), optical resolution constraints make it difficult to distinguish nuclear versus somatic photoconversion. As a result, photoconversion intended for the nucleus often inadvertently activates the soma-localized PCFPs.

Following nuclear photoconversion, PCFPs redistribute via passive diffusion through nuclear pores into the cytoplasm and eventually into neurites. While diffusion into large-diameter processes like axons may occur relatively quickly, labeling of thin and distal neurites is considerably slower due to their small cross-sectional area and the limited pool of photoconverted protein (only a fraction of PCFPs reside in the nucleus).

Thus, effective labeling of fine neurites with conventional PCFPs typically requires multiple rounds of photoconversion, which prolongs experimental timelines and risks phototoxicity.

2. The advantages of Pisces

Pisces is engineered to localize exclusively to the nucleus prior to activation, leaving all processes unlabeled (colorless).

Upon light stimulation, Pisces undergoes activation, NES-mediated export from the nucleus to the cytoplasm and neurites. This targeted transport bypasses the reliance on passive diffusion, enabling rapid and uniform labeling of even thin neurites with a single round of photoconversion.

The nuclear confinement of Pisces prior to activation ensures high signal specificity and eliminates background fluorescence in processes, which is unachievable with freely diffusing PCFPs.

In summary, soma-localized photoconversion of freely diffusing PCFPs does not overcome the inherent limitations of passive diffusion. In contrast, Pisces' nuclear pre-localization and active export mechanism enables rapid, specific, and efficient neurite labeling unachievable with conventional PCFPs.

7. On lines 134-136, the authors state that the rapid diffusion is facilitated by the NES. The precise meaning of this statement is not clear to me, and it is also not clear if there is any evidence to support it. Are the authors suggesting that the diffusion is more rapid with an NES than without an NES? If so, is there a reasonable mechanistic explanation?

Thank you for raising this critical point. We acknowledge that the original phrasing ("rapid diffusion facilitated by the NES") might be imprecise and apologize for the confusion.

The nuclear export signal (NES) enables active, energy-dependent nuclear export via the well-established exportin-1/RanGTP pathway (Ullman et al., 1997). This process is fundamentally distinct from passive diffusion in two key aspects:

1. **Directionality:** NES-mediated transport is unidirectional, exporting proteins from the nucleus to the cytoplasm, while passive diffusion is inherently bidirectional and nonspecific.
2. **Efficiency:** Active export bypasses the thermodynamic and spatial limitations of passive diffusion, resulting in significantly faster and more directed protein redistribution.

To support this, we quantified the effective transport kinetics of Pisces along neurites and found that the redistribution rate is approximately 10-fold faster than rates previously reported for passive diffusion alone (Brown, 2000).

Moreover, passive diffusion is particularly inefficient in neurons due to geometric constraints, especially in thin, distal neurites where diffusion is limited by small cross-sectional areas and cytoskeletal barriers. NES-mediated export overcomes these limitations by enabling direct, active transport, facilitating rapid labeling even in the most distal compartments.

We have revised the relevant statements in the discussion section of the manuscript to more accurately reflect this mechanism and avoid misinterpretation.

The revision in the discussion section is as follows (lines 290-295).

Unlike traditional genetically encoded methods (see Extended Data Figure 2j), which exhibit limited effectiveness in ambient light conditions, require repeated activation in the soma region, and take days to weeks for complete labeling due to slow protein diffusion rate (~0.02 - 0.09 $\mu\text{m/s}$), Pisces achieves precise manipulation by nuclear localization and rapid active, energy-dependent nuclear export by NES (~1 $\mu\text{m/s}$) (Supplementary Table 1).

Reference

Ullman, K. S., Powers, M. A. & Forbes, D. J. Nuclear Export Receptors: From Importin to Exportin. *Cell* **90**, 967–970 (1997).

Brown, A. Slow axonal transport: stop and go traffic in the axon. *Nat. Rev. Mol. Cell Biol.* **1**, 153–156 (2000).

8. To discover Pisces0.1, the authors seem to have tested more than 30 different constructs (as listed in Extended Data Fig. 2c) to find the best combination. The exact number of constructs tested should probably be mentioned in the main text.

Thank you for this helpful suggestion. We agree that including the exact number of constructs tested provides important context regarding the scope of our screening and optimization efforts.

We have now updated the main text to state that a total of 35 constructs were designed and evaluated during the development of Pisces0.1 (as listed in Extended Data Fig. 2c). This addition highlights the engineering process that was essential to identifying the optimal design.

The revision is as follows (lines 99-100).

We examined the translocation efficiency of the 35 combinatory constructs in H1299 cell cultures using a custom-made LED setup.

9. Confocal is misspelled in all instances where it occurs inside a figure.

We sincerely appreciate your careful attention to detail. We apologize for the repeated misspelling of “confocal” in the figure panels. This error has now been corrected in all affected figures in the revised manuscript.

Reviewer #2 (Remarks to the Author):

Uncovering neuronal cell types through their morphology, cell body location, function, and transcription is a major focus in neuroscience and developmental biology. Therefore, establishing a method to bridge detailed cellular structures, neural function, and transcription at single-cell resolution is both important and challenging. Previous studies utilizing photoactivated GFP (paGFP) allowed for the visualization of morphology at the single-cell level following functional imaging with GCaMP. However, due to the similar emission wavelengths of GCaMP and paGFP, it was difficult to separate activated and non-activated neurons. Additionally, lack of precision in targeting the activation wavelength posed an issue. In the present study, the authors engineered a light-inducible color-changing molecule called Pisces, which does not overlap with GCaMP's excitation properties and is less prone to leakage due to its self-cleavage site. By combining these two features, they successfully labeled neurons within a short time window, which is critical for

labeling cells in a developing organism. They demonstrated that multiple neurons could be labeled sequentially, thereby nicely reconstructing neural structures.

Although the authors successfully demonstrated the ability to connect morphology and neural function at the single-cell level, the relationship between morphology and transcription is not presented in the same context; instead, only the transcriptional cell types within groups of neurons are shown. The authors provided a structural analysis of selected neurons in the habenula in Figure 3, and a functional and morphological analysis of selected neurons in the midbrain in Figure 5. In Figure 6, they labeled habenula and midbrain neurons broadly and sparsely in different fish, followed by single-cell RNA sequencing (RNA-seq). Consequently, we can only very roughly deduce the correspondence of cell types in the habenula, but not with the same precision in the midbrain. Notably, the resolution of cell types identified through functional and transcriptional analyses was quite different. While it is understandable that there are technical difficulties in matching single-cell RNA-seq to the morphotypes of neurons, the authors claim to integrate morphology and molecular information for individual neurons (as illustrated in Figure 6i). Therefore, it is necessary to demonstrate the correspondence between morphology and transcription. One suggestion is first classifying the neurons by morphology and transcription independently, and investigating if the two classifiers match or not.

We thank the reviewer #2 for highlighting the importance of integrating neuronal morphology and transcription at the single-cell level. We agree that our current study does not achieve full multimodal integration in individual neurons, and we appreciate the opportunity to clarify this limitation and describe ongoing efforts.

While Pisces enables single-cell morphological and functional analysis, several technical challenges currently prevent the integration of transcriptomic profiling in the same neurons:

1. Time Requirement for Morphological Tracing

Morphological reconstruction of single neurons using Pisces typically requires ~5 hours per neuron (4 hours post-photoactivation and approximately 1 hour for imaging). This significantly limits the throughput and feasibility of combining with transcriptomic profiling, which must occur promptly to preserve RNA integrity.

2. Low Throughput in Functional-Morphological Experiments

The use of specific transgenic lines constrains the number of neurons we can label per fish (usually 1~2) and the number of fish we can perform per day (usually <10). This limits the ability to collect sufficient morpho-functional data for subsequent single-cell RNA-seq.

3. Challenges in Single-Cell RNA-seq from Zebrafish Neurons

Zebrafish neurons typically contain low transcript levels, and the dissociation and library preparation steps (which take hours to days) often lead to suboptimal cDNA quality, hindering high-fidelity transcriptomic profiling at single-neuron resolution. These limitations have been clarified in the revised manuscript, and we have adjusted our claims accordingly.

To address this gap toward morphology-transcription mapping, we performed a customized multiplexed fluorescence in situ hybridization (easiFISH) on *Pisces*-labeled individual habenular neurons in *Tg(elavl3:Gal4-VP16);Tg(UAS-E1B:Pisces)* fish. Traced neurons from two groups of fish were probed for *gng8* and *tac3a*, respectively (n = 3 fish for each group). Most traced neurons expressed *gng8*, while *tac3a* was restricted to right-sided neurons. These results (Fig. R2.1, Fig. 6f, Extended Data Fig. 7e, Supplementary Video 12 and 13) suggest a single-neuron level correlation between molecular identity and morphotype, though we emphasize that this does not yet demonstrate single-neuron multimodal integration including functions, morphologies and molecules. As suggested, we plan to independently classify neurons by morphology and transcription in future work and examine their correspondence. Improving transgenic tools, labeling throughput, and integrating spatial transcriptomic methods will be key to achieving full single-cell resolution across modalities.

Fig. R2.1a,b. Representative z-axis maximum projection images of all slices illustrating the integration of mRNA spatial localization and single habenular neuron morphology in zebrafish

larvae expressing neuronal Pisces. Left: Morphology of individual habenular neurons imaging before easiFISH. Zoomed-in view of the activated neuron (dashed square) is shown on the middle of this panel. Middle: Z-axis maximum projection images of 10 slices containing activated Pisces spatial expression patterns of *gng8* and *tac3a* mRNAs detected using easiFISH. Nuclear-localized Pisces, activated neurons, and mRNA signals are shown in green, red, and white, respectively. After performing easiFISH, the fluorescence signal of the neurites of the activated neuron diminished, leaving only the fluorescence signal from the nucleus visible. Right: Morphological reconstructions of individual neurons based on images before easiFISH. The activated neurons are indicated by blue arrows. Larvae were raised under ALE conditions. Scale bars: 20 μm .

Additionally, the following are some other points needed to be described or considered in the manuscript,

(1) Spatial limitation of visualization:

The manuscript only presents data on neurons located in the dorsal part of the brain. This raises concerns regarding the completeness of the analysis, as the ventral part is not represented. Please clarify if there are inherent limitations in the visualization technique that preclude the examination of neurons in this region. A more comprehensive exploration of both dorsal and ventral neurons would enhance the study's impact.

We thank the reviewer for this thoughtful comment. The majority of the representative examples in the manuscript are from dorsal (tectal neurons) and medium (LC/NE neuron) brain regions because these neurons yield stronger fluorescence signals and higher signal-to-noise ratios (SNR) under our confocal imaging conditions. These make them more suitable for demonstrating the resolution and morphological clarity achievable with the Pisces system.

To address the reviewer's concern regarding ventral brain regions, we conducted additional experiments specifically targeting neurons with somata located at deeper positions (180-220 μm below the dorsal surface, $n = 3$ fish). We have included a representative example of a reconstructed ventral neuron in the revised main text (Fig. R2.2, Extended Data Fig. 4c and Supplementary Video 5), along with updated descriptions in the Results. Our results confirmed that Pisces is capable of labeling and reconstructing individual neurons in ventral areas, although the fluorescence intensity and SNR were lower due to optical scattering and light attenuation at increased imaging depths, a known limitation of confocal microscopy in thick biological tissues.

Importantly, the ability of Pisces to label ventral neurons demonstrates that the system itself does not impose a spatial restriction. The limitation lies primarily in the imaging method rather than in the labeling strategy. These changes clarify the applicability of Pisces across the entire brain and more fully address the completeness of our analysis.

The revisions are as follows (lines 155-161):

To assess the ability of Pisces to label the morphology of neurons in ventral brain regions, we targeted neurons with somata located in deeper regions (~180 - 220 μm below the dorsal surface). Pisces is capable of labeling and reconstructing individual neurons in these ventral areas (**Extended Data Fig. 4c and Supplementary Video 5**). However, the fluorescence intensity and signal-to-noise ratio (SNR) of ventral neurons were lower than those of dorsal neurons, primarily due to optical scattering and light attenuation at greater imaging depths, a known limitation of confocal microscopy in thick tissues.

Fig. R2.2. Representative z-axis maximum projection images of all slices depicting the morphology of two ventral neurons, labeled as neuron 1 and neuron 2. The depths of two neurons from the dorsal side are $\sim 180 \mu\text{m}$ and $\sim 220 \mu\text{m}$, respectively. Zoomed-in views of the activated neurons (indicated by the dashed square) are displayed in the lower left and lower right corners of this panel. The blue arrows indicate the neurons specifically activated for morphological tracing, while the white arrows highlight neurons with mMaple activation but without PhoCl activation, resulting in red nuclei. The corresponding morphological traces of

these neurons are shown right. The larvae were raised under ALE conditions. Scale bars: 50 μm for full images, 10 μm for zoomed-in views.

(2) Homogeneity of expression:

The expression levels of the Pisces protein across neuronal populations appear to be homogeneous, yet the current data lacks a quantifiable measure. Since the fish examined are first-generation injected fish, it is plausible that expression varies among neurons. I recommend providing a quantification of the percentage of neurons expressing the plasmid to substantiate claims regarding the technique's efficiency and robustness.

We thank the reviewer for raising this important point. Indeed, in our initial experiments with first-generation injected (F0) fish, we observed relatively consistent nuclear-localized fluorescence across many neurons, likely due to the robust expression driven by the UAS element in the Gal4 system. Still, we agree that quantitative data is necessary to support claims of robustness and efficiency.

In response to this comment and to provide more controlled expression variability, we leveraged established transgenic fishline *Tg(elavl3:Gal4-VP16);Tg(UAS-E1B:Pisces)* with varying labeling densities ranging from 7,000 to 40,000 neurons, among a total of \sim 100,000 neurons expressed Pisces in the larval zebrafish brain. Our results show that Pisces effectively labels and reconstructs the morphology of individual neurons in zebrafish with different densities ($n = 3$ fish). As expected, denser expression demands more precise photoconversion targeting, increases the risk of off-target activation, and requires more time to obtain clear images by eliminating noise from undesired mMaple signals in the nucleus of adjacent neurons. Despite these challenges, we were able to achieve specific single-cell labeling even in the dense-expression group with careful experimental handling.

To reflect these findings, we have included a quantification of labeled neuron density and corresponding examples in the revised manuscript (Fig. R2.3a,b, Extended Data Fig. 4d,e and Supplementary Video 6), and we have revised the Results and Discussion sections accordingly. These additions better substantiate the versatility and robustness of the Pisces system across different expression levels.

The revisions are as follows (lines 162-168):

Tracing the morphology of individual neurons in zebrafish with dense Pisces expression presents a significant challenge. To evaluate Pisces' ability

to label neuronal morphology in larvae with varying densities of Pisces-positive neurons, we conducted neuronal labeling in the *Tg(elavl3:Gal4-VP16);Tg(UAS-E1B:Pisces)* larvae, which exhibited different labeling densities in the larval zebrafish brain. With careful experimental handling (see **Methods**), *Pisces* effectively labeled and reconstructed individual neuronal morphologies, even in brains with high density of *Pisces* expression, reaching up to 40% (**Extended Data Fig. 4d,e and Supplementary Video 6**).

Fig. R2.3a,b. Representative z-axis maximum projection images of all slices showing the morphology of activated neurons in zebrafish larvae with moderate expression (~27%, a) and high expression (~40%, b). In panel b, two activated neurons are labeled as neuron 1 and neuron 2. The zoomed-in views of the activated neurons, marked by the dashed square, can be found in the lower corners of each panel. The neurons activated specifically for morphological tracing are indicated by blue arrows, while white arrows point to neurons exhibiting mMaple activation but lacking PhoCl activation. The corresponding morphological traces of these neurons are shown right. The larvae were raised under ALE conditions. Scale bars: 50 μm for full images, 10 μm for zoomed-in views.

(3) Registration quality:

The quality of registration for imaging stacks is crucial for accurately interpreting neuronal structure and connectivity. The methods section does not adequately describe the registration approach employed in this study. Given the incomplete expression of *Pisces* in the larvae depicted in Figures 1, 2b, 3a, and 4, I encourage the authors to provide more details on the registration process and to ensure that the positional accuracy of cell bodies and projections is addressed explicitly.

We thank the reviewer for highlighting the importance of registration accuracy in interpreting neuronal structure and connectivity, particularly in the context of incomplete marker expression. The detailed methods is described as reported (Du et al., 2025). We used the open-source Advanced Normalization Tools (ANTs) to perform high-precision, multi-step registration of our imaging datasets. Our pipeline includes rigid and affine (RA) transformations followed by deformable (SyN-based warp) registration.

The integrative calcium imaging and morphology dataset (Fig. 5 and Extended Data Fig. 5) were obtained from *Tg(elav13:GCaMP6s);Tg(elav13:Gal4-VP16);Tg(UAS:Pisces)* larvae, with pan-neuronal expression of cytosol-localized GCaMP6s and mosaic expression of nucleus-localized *Pisces*. The *Tg(elav13:GCaMP6s)* expression pattern was used as the reference channel, and data were registered to the *Tg(elav13:NES-jRGECO1a)* template brain (Du et al., 2025). For analyzing dendritic arborization of tectal neurons within specific neuropil layers, we used the *Ki(isl2b:Gal4FF)* line with sparse *UAS:tdTomato-CAAX* expression to obtain the layered optic tectum subregions based on retinal ganglion cell (RGC) projections.

For the morphology datasets obtained from larvae with mosaic expression of nucleus-localized *Pisces* (Figs. 3, 4, 6, Extended Data Figs. 3-5, 7), the presented traces were reconstructed within respective imaging spaces and were not registered. We previously tried registration based on brain contours, but the registration accuracy was poor.

To evaluate the positional accuracy of registered cell bodies and projections, we used two complementary metrics:

1. **Mean Landmark Distance (MLD):** We annotated 13 anatomical landmarks across the brain in both individual scans and templates. These were verified independently by three annotators. After applying the full registration pipeline, we computed the Euclidean distances between corresponding landmarks, yielding a mean positional deviation of $3.10 \pm 1.10 \mu\text{m}$. For the single-cell reconstruction dataset specifically, we validated 88 scans and found consistent accuracy ($\text{MLD} = 3.28 \pm 0.06 \mu\text{m}$).

2. **Normalized Cross-Correlation (NCC):** As an additional automated measure, we assessed image similarity at the rigid/affine registration step and proceeded to warp registration for datasets with $NCC > 0.55$, a threshold empirically selected based on manual inspection.

Together, these strategies ensure that despite incomplete Pisces expression in some samples, cell body and projection locations are preserved with subcellular precision during registration.

We have now revised the Methods section as follows (lines 505-522):

Single-neuron morphologies were semi-automatically reconstructed using the Simple Neurite Tracer plugin in ImageJ Fiji, as previously described. Tracing began at the soma and extended outward in high-resolution NRRD images centered on the cell body. The averaged morphology stack from each time series was aligned to a 6-dpf Tg(HuC:H2B-GCaMP6f) template brain (with annotated regions) using Symmetric Normalization (SyNRA) in Advanced Normalization Tools (ANTsPyX v0.3.728)59,60 with default parameters on Ubuntu 20.04.5. To analyze layer-specific neurite distribution, we used the Ki(isl2b:Gal4FF);UAS:tdTomato-CAAX line to establish a layered OT subregions based on retinal ganglion cell (RGC) projections.

Neuron morphologies and brain region data were exported in SWC and STL formats. Projection intensity was estimated by quantifying total fiber length within segmented regions and visualized with custom 3D software. A dual-channel reference template was generated from elavl3:H2B-GCaMP6s;vglut2a:DsRed larvae. Confocal stacks were resampled to isotropic resolution and registered via shape-based averaging and deformable registration using ANTs. Rigid and affine alignment was followed by iterative deformable warping, with convergence assessed via normalized cross-correlation (NCC) scores. The registration pipeline included two parallelized steps: rigid/affine registration, followed by deformable warping. Background fluorescence was excluded using a fish-body mask generated via the Computational Morphometry Toolkit (CMTK) and manually refined.

Reference

Du, X.-F. *et al.* Central nervous system atlas of larval zebrafish constructed using the morphology of single excitatory and inhibitory neurons. *bioRxiv* 2025.06.06.658008 (2025) doi:10.1101/2025.06.06.658008.

(4) Development of the larval brain during labeling:

The manuscript should consider discussing the developmental changes occurring in the larval brain over the labeling period provided. It is important to address potential effects of cell division or the alteration of neural projections that could influence results at different ages. In addition, whether transcription changes before and after the labeling is critical to be mentioned. Clarifying the applicability of the Pisces technique across various temporal scales would greatly enhance the robustness of the findings.

Thank you for raising this important point. In our current study, we used the Pisces system to monitor neuronal dynamics in the zebrafish larval brain at 3 days post-fertilization (dpf) over a 14-hour imaging period. During this time, we observed morphological changes such as filopodia trimming and somata-neurite dynamics in tectal neurons. However, we did not detect any neurogenesis events, such as cell division, within the labeled neuronal population during the imaging session.

We acknowledge that the larval brain continues to develop at this stage, and that extended imaging sessions could be influenced by transcriptional changes or remodeling of neural projections. Although we did not observe cell division in the labeled neurons, we agree that such events may occur outside of our field of view or in other cell populations. Nevertheless, the stability and completeness of Pisces labeling over extended periods supports reliable readouts of neuronal development.

Transcriptional dynamics are also a critical consideration. While Pisces currently provides high-resolution morphological information, it does not directly capture gene expression changes. In future work, integrating Pisces with single-cell or spatial transcriptomics will allow us to better link morphological dynamics with molecular information.

Regarding photoconversion, we used a weak 405 nm laser (0.15-0.5 μ W) for a short duration (10-60 s), which is significantly milder than typical imaging conditions. Based on previous studies and our own observations, we do not expect this level of light exposure to significantly affect transcriptional activity.

In summary, while Pisces enables long-term, high-resolution morphological tracking, we agree that developmental and transcriptional dynamics during the imaging period are important factors. Future integration with single-cell multi-modal atlases will help address these aspects and enhance the temporal robustness and broader applicability of Pisces across developmental stages.

The revisions are as follows (lines 313-316):

Given the conservation of neural developmental programs across vertebrates, Pisces holds significant potential for applications in developmental neuroscience, such as the construct of a comprehensive neuronal developmental landscape to link morphological dynamics with molecular information.

Minor corrections in figures:

(1) Figure 3a: In the leftmost schematic drawing of the fish larvae, the location of the green blobs appears to correspond to the olfactory bulb rather than the habenula, which is misleading. This issue is also present in Extended Data Fig. 3.

Thank you for pointing this out. We agree that the current illustration is misleading. We revised the schematic in Fig. 3a to accurately reflect the anatomical location of the habenula. Corresponding adjustments have also been made in Extended Data Fig. 3 to ensure consistency across the figures.

(2) Figure 2: Figure 2 contains two instances of "b" and no instance of "c."

We appreciate your careful review. This was an oversight on our part. We have corrected the labeling in Fig. 2 to maintain proper panel designation and clarity.

(3) Figure 5f: I observe a few projections that seem to cross between the two hemispheres, yet they are labeled as the ipsilateral projection dataset. Could you please clarify this?

We appreciate the reviewer's careful observation. While the dataset presented in Fig. 5f focuses on neurons with predominantly ipsilateral projections, we acknowledge that a small subset (2 neurons in ipsilateral type) of axonal branches appears to extend slightly across the midline but do not reach the contralateral OT. However, the majority of these neurons exhibit neurite arborization confined to the same hemisphere, and their main descending projections target the ipsilateral hindbrain. Therefore, we classified them as ipsilateral projection neurons.

Importantly, our classification of ipsilateral versus contralateral projection types is based primarily on the trajectory of their long-range descending axons, particularly those extending from the midbrain to the hindbrain. While some neurons do cross the midline anteriorly, such as through the optic chiasm or proximal axon bifurcations, these do not represent descending projections and are not considered indicative of contralateral hindbrain targeting.

We have now clarified this distinction in the revised figure legend to avoid confusion.

The revisions are as follows (lines 860-863).

A small subset (2 ipsilateral neurons) shows axonal branches crossing the midline, but they do not reach the contralateral OT. The majority, however, have neurite arborization confined to the same hemisphere, with their main projections targeting the ipsilateral hindbrain, thus classifying them as ipsilateral projection neurons.

Questions in the videos:

Video1: Since the ventral part of the brain does not express signals, it is not convincing that the cellular morphology of neurons in the whole brain is trackable or not. If there is a spatial limitation in the technique, then it should be mentioned in the manuscript. It would be great if an overlay of the schematic drawing of the whole brain was shown with the imaging stacks.

We thank the reviewer for this insightful observation. The apparent absence of signal in the ventral brain is not due to a lack of Pisces expression, but rather to limitations of our imaging protocol. Specifically, we performed high-resolution confocal imaging on the central ~40 optical slices, which primarily capture dorsal and mid-regions. Fluorescence signal diminishes significantly as imaging depth increases due to light scattering and reduced photon collection efficiency, thus it is an inherent limitation of confocal microscopy in thick tissues.

This does not reflect a spatial constraint of the Pisces system itself, as ventral neurons are indeed transfected and capable of photoconversion in Fig. R2.1. To address the reviewer's concern, we have updated the corresponding figure legend to explicitly describe the imaging depth and protocol.

It would be beneficial to include a 3D schematic drawing of the whole brain with the imaging stacks; however, there is a technical challenge for us in doing so and we aim to further develop the skills in future work.

The revisions are as follows (lines 1150-1153).

High-resolution confocal imaging was conducted on approximately 40 central optical sections of larval zebrafish, primarily capturing dorsal and mid-regional neurons. While ventral neurons were also activated and traced by Pisces during the experiment, they were not included in the imaging field due to the protocol.

Video2: The manuscript said the single neurons are labeled, but I can see two blobs, which should be the cell bodies, in each hemisphere. Is this an artifact of imaging or two cells are visualized in each hemisphere? And if it's the latter, then it's hard to say this is a single-cell projection.

We appreciate the reviewer's careful attention to this detail. The two fluorescent somata observed in each hemisphere in this video do not represent dual-neuron labeling (Fig. R2.4 and Fig. 3c). Specifically, these signals arise from: (1) a fully labeled habenular neuron, and (2) an adjacent neuron with partial nuclear-localized mMaple activation. This optical configuration (3D reconstruction at identical contrast settings) simultaneously shows both neuronal somata and their extending neurites, which may create the illusion of dual labeling.

Fig. R2.4. Representative z-axis maximum projection images of the morphological projections of two activated habenula neurons on both sides (c) on 6-dpf larval zebrafish. The brightness of red fluorescence was adjusted to visualize the nuclei of activated neurons in the lower panels. Numbers and white arrowheads indicate the activated neurons. Morphological traces of each neuron are displayed on the right ($n = 3$ fish). Scale bars: $50 \mu\text{m}$ in whole images, $10 \mu\text{m}$ in zoomed views. Hb, habenula; IPN, interpeduncular nucleus. Larvae were raised in constant darkness.

Pisces relies on simultaneous photoconversion of both PhoCl and nuclear-localized mMaple to enable neurite labeling. However, due to the use of a 405 nm laser during photoconversion, nearby neurons can exhibit partial activation of nuclear mMaple, leading to faint somatic fluorescence in non-target cells. Crucially, because PhoCl is not activated in these adjacent neurons, they do not exhibit neurite labeling.

The visibility of the second faint signal is a consequence of post-acquisition contrast enhancement applied to better visualize weak axonal projections. We have revised the Supplementary Video 2 figure legend to explicitly clarify this distinction.

The revisions are as follows (lines 1157-1161).

Note that the two fluorescent somata observed in each hemisphere in this video do not represent dual-neuron labeling (Fig. 3c). Specifically, these signals arise from: (1) a fully labeled habenular neuron, and (2) an adjacent neuron with partial nuclear-localized mMaple activation (Fig. 3c). This optical configuration (3D reconstruction at identical contrast settings) simultaneously shows both neuronal somata and their extending neurites, which may create the illusion of dual labeling.

Video3: It would be more convincing if the signals are overlaid with the skeletons of the tracked projections.

We fully agree with the reviewer that overlaying the tracked projections with their corresponding skeletonized reconstructions enhances the clarity and interpretability of the neuronal tracing. In response, we have updated Video 3 (now Video 4) by adding a new video that overlays the fluorescent signals captured by Pisces with the skeletons of the traced projections. In most regions, the overlays demonstrate a good correspondence between signal and skeleton. However, we note that in some areas, particularly where signal intensity is low, the fluorescent signal may appear faint or unclear in the full-frame image.

We have added this point into Video legends (lines 1174-1177).

In most regions, the overlays show strong correspondence between the Pisces signals and the skeletons of the traced projections. However, we note that in some areas, particularly where signal intensity is low, the fluorescent signal may appear faint or unclear when contrasted with the full-frame image.

Reviewer #3 (Remarks to the Author):

Rong-Kun and co-authors presents a genuinely elegant manuscript describing a new approach to photo-label single neurons in-vivo, a method they termed Pisces. The methods allows to acquire functional, morphological and transcriptomic data on a single neuron level. I am certain that the method has the potential to dramatically push forward neuronal-circuits studies.

I would like to point to two fundamental issues that require in-depth revision of the current manuscript.

3.1 First and most important to me, is the fact that the authors claim that the labeling is "complete" neurons (e.g. [... capability of Pisces in outlining the complete morphology...This demonstrates Pisces' fidelity in tracing entire neuronal morphologies... pp5 lines 122-9] OR [... we demonstrate that Pisces enables high-resolution and clear tracing of the complete morphology of multiple neurons... , pp10, line 265-6]. The data provided to support this claim (e.g. Extended Data Fig 3) is not sufficient for this reviewer to appreciate the completeness of the photo-activated labeling. Such a bold statement has to be accompanied with a clear comparison with labeling the same neurons with an independent technique of their choice.

We sincerely appreciate the reviewer's rigorous scrutiny of this critical issue. To substantiate our claims, we have conducted additional validation experiments and revised the manuscript accordingly:

1. Independent validation through dual-reporter co-labeling

Using sparse labeling via the established single-cell electroporation method with Alexa Fluor 488 and Pisces, we observed the spatial colocalization between the photoconverted Pisces signal and the Alexa Fluor 488-labeled neuronal outlines (Fig. R3.1a,b, Extended Data Fig. 4a,b and Supplementary Video 3). Pisces accurately captured the full extent of neuronal morphology, showing no missing distal processes when compared to the widely used single-cell electroporation technique (n = 3 fish). Notably, Pisces labeled additional neuronal processes that Alexa Fluor 488 failed to capture (Fig. R3.1b, black arrows). Furthermore, the single-cell electroporation method exhibited nonspecific labeling of adjacent neurons in regions of tight junctions (Fig. R3.1b, white arrows), likely due to diffusion of the small Alexa Fluor 488 dye through connexin channels. These findings strongly support the accuracy and completeness of Pisces-mediated neuronal labeling.

2. Established benchmarks in habenular circuitry

In addition, the habenular-to-IPN projection in zebrafish is a well-characterized and stereotyped pathway. As shown in Fig. R3.2a-c (Extended Data Fig. 3a-c), Pisces-labeled habenular neurons consistently project to the interpeduncular nucleus (IPN), and time-lapse imaging over a 4-hour period revealed no morphological changes or delayed axonal labeling. This stability of labeled axons during post-conversion imaging further supports the idea that photoconversion captures the full extent of the neuron's existing morphology.

Taken together, these results, from both independent co-labeling and well-established neuroanatomical benchmarks, provide strong support for the fidelity and completeness of Pisces-mediated photolabeling.

The revisions are as follows (line 142-147).

We further validated the morphological completeness of Pisces signals by performing single-cell electroporation of Alexa Fluor 488 dye into Pisces-activated neurons in *Tg(elavl3:Gal4-VP16);Tg(UAS-E1B:Pisces)* zebrafish larvae, which exhibited sparse Pisces expression (Extended Data Fig. 4a,b and Supplementary Video 3). This experiment revealed colocalization of Pisces (red) and Alexa Fluor 488 (green) signals along axonal paths and terminals, demonstrating Pisces' reliability in tracing the complete morphology of individual neurons.

Fig. R3.1a,b Assessment of labeling completeness through colocalization of dual reporters. **a**, Schematic illustration of the experimental procedure to label the morphology of a single neuron in larval zebrafish at 6-dpf using nuclear Pisces and single-cell electroporation. Sparse labeling

by single-cell electroporation of Alexa Fluor 488 dye was performed after labeling the single neuronal morphology with Pisces under a single pulse of 405-nm laser (0.5 μ W for 60 s). **b**, Representative z-axis maximum projection images of illustrate colocalization of the same neuronal morphology labeled by single-cell electroporation using Alexa Fluor 488 and by Pisces. Blue arrows indicate the neuron activated for morphological tracing. White arrows highlight another neuron that was inadvertently labeled during single-cell electroporation. Black arrows in the right point to neurites that remained unlabeled due to limitations of the single-cell electroporation method. Larvae were raised in constant darkness or in ambient light environment (ALE). Scale bars: 20 μ m.

Fig. R3.2. Validation of labeling completeness through established benchmarks in habenular circuitry. **a.** Typical morphological projection of habenular neurons, with axonal projections extending through the fasciculus retroflexus (FR) fiber bundles to the interpeduncular nucleus (IPN). **b,c** Representative z-axis maximum projection images of habenular neurons within 4 hours in larval zebrafish raised in dark (**b**) and ALE (**c**) conditions. Full neuronal morphology was dimly labeled within 1 hour, with brightness increasing over 3 - 4 hours (upper panel). Dashed squares highlight zoomed-in views of the habenula (Hb, middle panel) and the IPN (bottom panel), were clearly visualized in both dark and ALE conditions. The brightness of the habenula was adjusted to improve the visibility of the activated neuron. Traces of the two

habenular neurons at 4 hours post-activation are shown on the right. Scale bars: 50 μm for full images, 10 μm for zoomed-in views. n = 3 fish.

3.2 Second, the authors demonstrate the ability to perform this approach in zebrafish, they suggest the method will be "compatible with other small animal models" albeit no support for such a claim is provided. Given the extensive use of rodents in Neuroscience research, it would be a high priority to provide evidence that this approach is indeed compatible in mammals.

We thank the reviewer for this important point. We agree that directly extrapolating the utility of Pisces from zebrafish to mammalian models without experimental validation may be an overstatement. We have revised the relevant text in the manuscript to more accurately reflect the current scope and future directions of our work.

Pisces currently functions optimally in zebrafish due to the transparency of the organism and the high optical accessibility of its nervous system. In contrast, mammalian models present significant challenges for photoconversion-based approaches, particularly due to:

- **Limited optical penetration** of 405 nm light in brain tissue (typically <50 μm in the mouse cortex, vs. >500 μm in larval zebrafish),
- **Relatively low brightness** of the mMaple fluorophore used in Pisces, which may not support sufficient signal-to-noise ratios of imaging neurites in mammalian brain tissue.

To address these limitations, we are currently developing a next-generation variant with improved fluorescence intensity (preliminary data suggest over a 10-fold increase). We plan to evaluate its performance in mammalian systems in subsequent studies.

Accordingly, we have revised the original statement to avoid overgeneralization (lines 305-306):

While Pisces is effective in larval zebrafish, it would deserve to test its compatibility in other small animal models in the future.

We believe this more accurately represents the current status of the method and our ongoing efforts to extend its applicability to mammalian models.

Reviewer #4 (Remarks to the Author):

Thank you for your comments.